# Learning and Transferring Physical Models through Derivatives

**Alessandro Trenta**                              *alessandro.trenta@phd.unipi.it*
*Department of Computer Science, University of Pisa*

**Andrea Cossu**                                   *andrea.cossu@di.unipi.it*
*Department of Computer Science, University of Pisa*

**Davide Bacciu**                                  *davide.bacciu@unipi.it*
*Department of Computer Science, University of Pisa*

**Reviewed on OpenReview:** *https://openreview.net/forum?id=IbBCDDeDF7*

## Abstract

We propose Derivative Learning (DERL), a supervised approach that models physical systems by learning their partial derivatives. We also leverage DERL to build physical models incrementally, by designing a distillation protocol that effectively transfers knowledge from a pre-trained model to a student one. We provide theoretical guarantees that DERL can learn the true physical system, being consistent with the underlying physical laws, even when using empirical derivatives. DERL outperforms state-of-the-art methods in generalizing an ODE to unseen initial conditions and a parametric PDE to unseen parameters. We also design a method based on DERL to transfer physical knowledge across models by extending them to new portions of the physical domain and a new range of PDE parameters. This introduces a new pipeline to build physical models incrementally in multiple stages.

## 1 Introduction

Machine Learning (ML) techniques have found great success in modeling dynamical and physical systems, including Partial Differential Equations (PDEs). The growing interest around this topic is driven by the many real-world problems that would benefit from accurate prediction of dynamical systems, such as weather prediction (Pathak et al., 2022), fluid modeling (Zhang et al., 2024), quantum mechanics (Mo et al., 2022), and molecular dynamics (Behler & Parrinello, 2007). These problems require grasping the essence of the system by modeling its evolution while following the underlying physical laws. Purely data-driven supervised models often fail at this task: while being able to approximate any function, they do not usually learn to maintain consistency with the physical dynamics of the system (Greydanus et al., 2019; Hansen et al., 2023), or fail to approximate it when only a few data points are available (Czarnecki et al., 2017). Physics-Informed Neural Networks (PINNs) (Raissi et al., 2019) emerged as an effective paradigm to learn the dynamics of a PDE, by explicitly including in the loss the PDE components evaluated using automatic differentiation (Baydin et al., 2018). This imposes physical consistency by design and allows the use of PINNs in regimes where data is scarce. Unfortunately, PINNs suffer from optimization issues (Wang et al., 2022a;b) which can lead to poor generalization (Wang et al., 2021).

We propose Derivative Learning (DERL), a new supervised approach to train neural networks using the partial derivatives of the objective function, as they perfectly describe its evolution in time and space (Section 3). Like PINNs, DERL also learns the initial and boundary conditions, as they are needed to retrieve the full solution. We formally prove that learning partial derivatives is sufficient to learn the evolution of the system, and we show how DERL achieves a better performance than state-of-the-art data-driven approaches on several physical systems, by testing its generalization on unseen data points and conditions (Section 4).

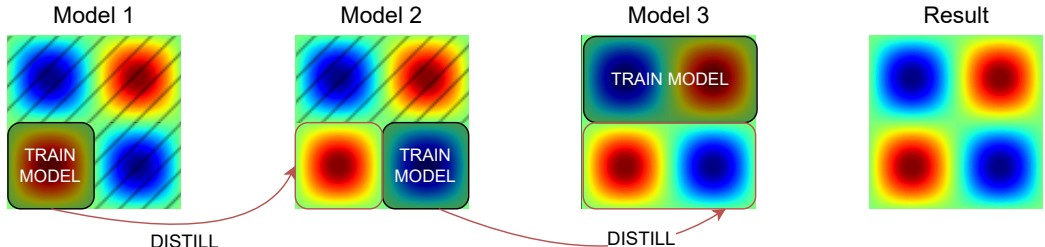

Figure 1: How to learn and transfer knowledge from a physical model. A physical model is first trained on a given domain. The model is not able to access the entire domain at once (the shaded area is unavailable). Later, the model is trained on a new portion of the domain, while previous knowledge is transferred and preserved through distillation. The same process can be applied to a time-space domain $[0, T] \times \Omega$. This way, DERL learns a solution incrementally in time. DERL implements both the learning and the transfer phase.

Physical systems are usually learned in isolation. However, the physical information acquired by a model on a given problem could be used by another model trained on a related one. We believe that the transfer of information across physical models deserves more attention and is currently understudied. The advantages of transferring information across models are well-known in machine learning. Nowadays, fine-tuning a pre-trained model on a domain-specific task instead of starting from scratch is the dominant paradigm.

Our approach is based on *distillation* (Hinton et al., 2015), where the physical knowledge contained in a fixed trained model (the teacher) is transferred to a student model by *distilling* the teacher's derivatives with DERL. We show that a distillation protocol based on DERL surpasses the performance of alternative protocols based on different physical information. We successfully transfer information to new portions of the domain, to new parameterization of the physical systems, and to longer time horizons (Section 5). In all cases, the resulting final model is able to solve the whole problem by only being trained on it in an incremental way. Figure 1 illustrates the aforementioned transfer protocol via distillation to new domains. To the best of our knowledge, this defines a new approach to transfer knowledge across physical models and to study their incremental design through distillation. More broadly, our transfer protocol can be applied in multiple stages, in a continual learning fashion (Parisi et al., 2019). In continual learning, a model is trained on a sequence of different tasks with the objective of learning new tasks without forgetting previous ones. Continual learning has never been studied on physical systems. Our experiments show that physical models exhibit some degree of forgetting, suggesting that applying existing continual learning strategies might help mitigate this issue on a long sequence of tasks. While we foresee the potential impact of continual learning for our case, in this paper, we focus on the transfer task through distillation (notably, a technique widely used also in continual learning (Li & Hoiem, 2016)) for 2 or 3 sequential steps.

Our main contributions are: (i) DERL, a new supervised learning approach to model physical systems. Section 3 and Figure 2 provide the technical description of DERL, while Section 4 reports our results on a variety of ODEs and PDEs against state-of-the-art methodologies. (ii) A theoretical validation of DERL which proves that the learned solution converges to the true one when the loss goes to zero (Section 3.1). Our results hold even with empirical derivatives when the analytical ones are not available. (iii) A distillation-based protocol powered by DERL, which is able to transfer physical knowledge from a reference model to a student one (Section 5). We discovered that distilling higher-order derivatives improves the final performance.

We hope our work can pave the way to an *incremental* and *compositional* way (Xiang et al., 2020; Soltoggio et al., 2024) of learning physical systems.

## 2 Background and Related Works

**Ordinary and Partial Differential Equations.** Many real-world physical systems are often described through Ordinary Differential Equations (ODEs), which can be written as

$$u^{(n)}(t) + a_1(t)u^{(n-1)}(t) + \ldots a_{n-1}(t)\dot{u}(t) + a_0 u(t) = f(t), \tag{1}$$

where $u^{(i)}(t) = \frac{d^i u}{dt^i}(t)$ is the $i$-th order derivative. The *order* of the ODE is defined as the highest order derivative in eq. (1), in this case $n$. An $n$-th order ODE can always be converted into a first-order system of ODEs by considering $\boldsymbol{u}(t) = [u(t), \dot{u}(t), \ldots u^{(n-1)}(t)]$, leading to

$$\dot{\boldsymbol{u}} = \boldsymbol{f}(\boldsymbol{u}(t), t; \xi), \qquad \boldsymbol{u}(0) = \boldsymbol{u}_0. \tag{2}$$

This definition actually allows for the representation of more complex systems, such as coupled oscillators. Here, $\boldsymbol{u} : [0, T] \to \mathbb{R}^n$ is a function describing the state of the system, $\boldsymbol{u}(0) = \boldsymbol{u}_0$ is its initial condition, and $\boldsymbol{f}(\boldsymbol{u}(t), t; \xi) : \mathbb{R}^n \to \mathbb{R}^n$ represents the dynamics and evolution of the system based on its current state and (possibly) parameters $\xi$. The existence and uniqueness of a solution are guaranteed under hypotheses such as Lipschitz continuity of $f$ (Hartman, 2002). Hence, eq. (1) indicates that the evolution of the physical system is completely described by its initial state and its derivatives.

PDEs describe physical systems on general domains $\Omega \subset \mathbb{R}^d$ with boundary $\partial\Omega$ through the conditions:

$$\begin{cases} \mathcal{F}[\boldsymbol{u}; \xi](t, \boldsymbol{x}) = 0 & t \in [0, T], \boldsymbol{x} \in \Omega, & \text{(PDE)} \\ \boldsymbol{u}(0, \boldsymbol{x}) = g(\boldsymbol{x}) & \boldsymbol{x} \in \Omega, & \text{(IC)} \\ \boldsymbol{u}(t, \boldsymbol{x}) = b(t, \boldsymbol{x}) & \boldsymbol{x} \in \partial\Omega, & \text{(BC)} \end{cases} \tag{3}$$

where $\mathcal{F}$ is a differential operator, $\xi \in \mathbb{R}^l$ is a set of parameters that regulates the dynamics, and $g(\boldsymbol{x})$ and $b(t, \boldsymbol{x})$ are functions that define the Initial condition (IC) and Boundary condition (BC), respectively. Problems of this kind are completely determined by the PDE, which describes the evolution and propagation of information from the IC and BC to the rest of the domain through time and space (Evans, 2022).

Classical numerical methods involve diverse strategies to solve ODEs and PDEs. While for the first, it might suffice to consider the integration in time $\boldsymbol{u}(t + \Delta t) = \boldsymbol{u}(t) + \Delta t \boldsymbol{f}(\boldsymbol{u}(t), t)$, PDEs often require more complex methodologies and assumptions to ensure the convergence to the true solution (Evans, 2022). A common strategy involves discretizing the variables on a grid to approximate derivatives, transforming the PDE into a system of ODEs (*finite differences*, Ferziger & Peric (2001)). Similarly, *finite volumes* methods (Ferziger & Peric, 2001) divide the space into small volumes, each one exchanging physical information and quantities with its neighbors via fluxes on their boundaries. Finally, the *finite element* method (Anderson et al., 2020) uses the weak formulation of PDEs, where the solution is tested against a base of functions on a mesh of points, transforming it into a complex system of ODEs. To apply classical methods, one needs to specify a pre-defined grid or mesh of points where the solution is calculated, which cannot be changed afterwards. Although being precise and providing theoretical guarantees on the convergence to the true solution as the grid becomes finer and finer, numerical methods are limited by being tied to a specific mesh. To predict the values of the solution for other points, they need to recalculate the entire solution or to rely on approximations and interpolations. Hence, they require accurately selecting the correct mesh, balancing refinement and complexity. In fact, they can also be very slow, as these methods need to invert quite large matrices. Hence, there has been a growing interest in ML models for PDEs as they are by far the fastest at inference and allow for easy queries of any point. Their capabilities are limited only by the availability of data and, in general, fewer results on theoretical convergence.

**Supervised learning methods.** Recurrent neural networks (Schmidt, 2019) are a prototypical example of systems that evolve through time based on their previous state, which makes them a good candidate to learn solutions to ODEs. However, they only work iteratively and are prone to error accumulation or the vanishing gradient effect (Pascanu et al., 2013). Similarly, ResNets (He et al., 2016) and NODEs (Chen et al., 2019) model the residual state update as a discrete or continuous derivative, but cannot predict complete trajectories at once or require an external ODE solver. Normalizing flows (Rezende & Mohamed, 2015) model dynamical systems such as particles by learning their distribution with iterative invertible mappings. Neural operators (Li et al., 2020b) and their evolution (Li et al., 2021) act as neural networks for entire functions by mapping initial conditions or parameters to a solution via kernel operators and Fourier transforms. While these methods are powerful and find many applications to real-world problems (Pathak et al., 2022), they require a large amount of data to generalize, are computationally intensive, and do not ensure that the solution is physically consistent. Neural Operators are also suited for a different purpose than ours, as they are not made to map a point in time-space $t, \boldsymbol{x}$ to its value $\boldsymbol{u}(t, \boldsymbol{x})$. Our work shares similarities with Sobolev

learning (Czarnecki et al., 2017; Srinivas & Fleuret, 2018), which adds derivative learning terms to the supervised loss with application to reinforcement learning and machine vision. As Sobolev learning is not designed for physical systems, we adapted it by adding the IC and BC into its loss, including a term for both $\boldsymbol{u}$ and its derivatives $\mathrm{D}u$. We show that the pure derivative approach of DERL achieves a better performance.

**Physics-inspired and Physics-Informed methods.** PINNs (Raissi et al., 2019) incorporate PDEs that describe the underlying physical system directly into the loss. Using automatic differentiation (Baydin et al., 2018), partial derivatives of the network are calculated on a set of points in the domain, called *collocation points*, and used to evaluate the PDE on the model $\mathcal{F}[\hat{u};\xi]$ to optimize it. Formally, given points $\{(t_i,\boldsymbol{x}_i)\}_{i=1,\dots,N_d}$ in the domain, the PDE residual loss is calculated by

$$\|\mathcal{F}[\hat{u}]\|_{L^2(\Omega)}^2 = \frac{1}{N_d}\sum_{i=1}^{N_d}\left(\mathcal{F}[\hat{u}(\boldsymbol{x}_i)]\right)^2, \tag{4}$$

where $\mathcal{F}[\hat{u}]$ is the evaluation of the PDE in equation 3 on the network $\hat{u}$. PINNs are used to increase the physical consistency of a model. However, PINNs suffer from optimization problems (Wang et al., 2021) and can fail to reach a minimum of the loss (Sun et al., 2020). To alleviate these issues, there exist methodologies to simplify the objective (Sun et al., 2020) or to choose the collocation points where the PDE residual is evaluated (Zhao, 2021; Lau et al., 2024). Both approaches are problem-specific and add complexity to the optimization process. Specific architectures for parametric PDEs are also developed (Zou & Karniadakis, 2023; Penwarden et al., 2023) but often require multiple instances to be trained. Sobolev norms have also been applied to PINNs for improving convergence guarantees (Son et al., 2021). Models inspired by physics exploit the formalisms of Hamiltonian (Greydanus et al., 2019) and Lagrangian (Cranmer et al., 2019) mechanics to be inherently consistent with the properties of the system, but they require the system to be conservative and to be (implicitly) formulated in the Lagrangian or Hamiltonian formalism, which is generally not the case. They also require an external solver to compute the trajectories.

**Knowledge transfer.** Distillation Hinton et al. (2015) was originally proposed as a method to transfer knowledge from a large teacher model to a small student model with minimal performance loss. Over time, distillation proved to be a useful tool for many cases Gu et al. (2023). Continual learning is one of them, where distillation is used to avoid forgetting by transferring the knowledge from a teacher trained on previous tasks to a student trained on the current one Li & Hoiem (2016); Carta et al. (2024). Existing reviews on knowledge transfer include (Gou et al., 2021; Yang et al., 2024) for an extensive treatment. We employ distillation as a means of enabling knowledge transfer across physical models.

## 3 Derivative Learning (DERL)

Models that learn dynamical systems in a supervised manner must be capable of simulating the evolution of the system in regions or at resolutions that are not available during training. Similarly, models for physical systems described by parametric equations are required to generalize to unseen sets of parameters. A model that is not consistent with the underlying physical laws will not provide reliable predictions under such conditions. Concretely, given a set of collocation points $\{\boldsymbol{x}_i\}_{i=1,\dots,N}$, $\boldsymbol{x}_i \in \mathbb{R}^d$ and their evaluation $\{u(\boldsymbol{x}_i)\}_{i=1,\dots,N}$ computed according to the true function $u : \mathbb{R}^n \to \mathbb{R}$, there exist infinite solutions that pass through those points. We are interested in learning *the one solution* that is compatible with the underlying physical model. In the theory of PDEs, uniqueness results require considering functional norms and types of convergence that go beyond the $L^2(\Omega)$ norm, related to the Mean Squared Error in ML. In particular, we need to consider the Sobolev space (Maz'ya, 2011), that is, the space of functions equipped with the norm

$$\|\boldsymbol{u}\|_{W^{m,2}(\Omega)}^2 = \|\boldsymbol{u}\|_{L^2(\Omega)}^2 + \sum_{l=1}^{m}\left\|\mathrm{D}^l\boldsymbol{u}\right\|_{L^2(\Omega)}^2, \tag{5}$$

where $\mathrm{D}^l\boldsymbol{u}$ is the $l$-th order differential, the set of $l$-th order partial derivatives. Sobolev learning (SOB, Czarnecki et al. (2017)) employs this norm to better train ML models in settings with scarce data applied to

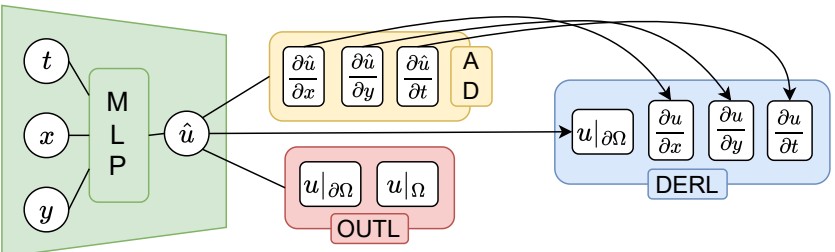

Figure 2: Graphical representation on how DERL learns a function $u(t, x, y)$. Partial derivatives $\frac{\partial \hat{u}}{\partial t}, \frac{\partial \hat{u}}{\partial x}, \frac{\partial \hat{u}}{\partial y}$ of the Neural Network (NN) output $\hat{u}$ are computed by Automatic Differentiation (AD). DERL *learns* the partial derivatives as independent targets, together with the initial or boundary condition $u|_{\partial\Omega}$. We compare DERL to OUTL, which learns the function $u$ directly in the domain $\Omega$ and on the boundary $\partial\Omega$.

reinforcement learning and machine vision. In the following Sections, we show that, instead, it is sufficient to consider only the derivative terms in eq. (5), leading to the same theoretical guarantees and a new universality theorem, but stronger experimental results due to a simpler loss. Finally, we also describe a new incremental paradigm to learn physical systems, employing our methodology, which alleviates PINN optimization issues.

We now introduce DERL (fig. 2), a supervised approach that learns from continuous dynamical systems such as ODEs and PDEs. The popular PINN model (Raissi et al., 2019) finds the solution to a PDE by minimizing its residual $\|\mathcal{F}[\hat{u}]\|^2$ computed on the model, which is taken as a measure of its physical consistency. DERL takes a different route. The key idea stems from the fact that the initial state and the state derivatives of a dynamical system are *necessary and sufficient* to predict its full trajectory. From a mathematical point of view, two functions with the same continuous derivative and with a point in common must coincide, in the spirit of the Fundamental Theorem of Calculus. With this in mind, DERL minimizes the following loss function:

$$L(\hat{\boldsymbol{u}}, \boldsymbol{u}) = \overbrace{\lambda_u \left\| \mathrm{D}\hat{\boldsymbol{u}}(t, \boldsymbol{x}) - \mathrm{D}\boldsymbol{u}(t, \boldsymbol{x}) \right\|_2^2}^{\textbf{Derivative learning}} + \overbrace{\lambda_B \left\| \hat{\boldsymbol{u}}(t, \boldsymbol{x}) - b(t, \boldsymbol{x}) \right\|_2^2}^{\text{Boundary cond.}} + \overbrace{\lambda_I \left\| \hat{\boldsymbol{u}}(0, \boldsymbol{x}) - g(\boldsymbol{x}) \right\|_2^2}^{\text{Initial cond.}}. \tag{6}$$

DERL's loss combines the $L^2$ loss on the function jacobians $\mathrm{D}\boldsymbol{u}$, the BC and the IC, where $\lambda_u, \lambda_B, \lambda_I$ are hyperparameters controlling the weight of each component. The first term corresponds to the $L^2$ loss on the gradient $\nabla u$ when $u(\boldsymbol{x}) \in \mathbb{R}$. The $L^2$ losses are computed empirically as the Mean Squared Error over a set of collocation points (Appendix A). In the case of time-independent problems, such as the Allen-Cahn equation (AC) in table 1, the last term of the loss is dropped. We implement $\hat{\boldsymbol{u}}$ as a multi-layer perception (MLP). DERL works even when analytical derivatives are not available, as empirical derivatives obtained with finite differences on $u$ (Anderson et al., 2020) are sufficient to train the network. Compared to a supervised approach that learns $u$ (called OUTL in the rest of this paper, fig. 2), we will show that DERL generalizes better to unseen portions of the input domain and to unseen PDE parameterization.

**Derivative distillation for transfer of physical knowledge.** We leverage DERL and distillation to transfer the physical knowledge contained in a pre-trained model to a randomly initialized student or to fine-tune the teacher itself (similar to continual learning) while preserving previous knowledge. To achieve so, we compute the (frozen) teacher derivatives $\mathrm{D}\hat{u}_T$ and the student derivatives $\mathrm{D}\hat{u}_S$ in the domain. The student network is trained with the loss:

$$L_{\text{student}} = L_{\text{system}} + \|\mathrm{D}\hat{u}_T - \mathrm{D}\hat{u}_S\|_2, \tag{7}$$

which comprehends both the loss on current data coming from the environment ($L_{\text{system}}$) *and* the distillation loss between teacher derivatives and the student derivatives. This relieves the burden of learning exclusively from raw data and it enables continual exchange of physical information across models. We will show that other distillation losses underperform with respect to DERL. The MSE replaces the KL divergence commonly used in distillation for classification problems. We will show that *distilling higher-order derivatives* leads to substantial improvements in the physical consistency of the final model.

We instantiate our incremental distillation protocol in three different configurations: i) by transferring knowledge to longer time horizons (Section 5.1), ii) to new portions of the domain (Section 5.2), and iii) to new parameterization of the physical system (Section 5.3). More in general, given $\Omega$ the whole space considered (domain, physical parameters, time), we partition it into non-overlapping subsets $\Omega = \cup_i \Omega_i$. We start by training a randomly-initialized PINN on $\Omega_1$, following the common training protocol described by Raissi et al. (2019). Then, for each step $i > 1$, we learn the new problem by minimizing a PINN loss on the new region $\Omega_i$ and we include previous knowledge through the distillation loss $\|D\hat{u}_T - D\hat{u}_S\|_2$ computed on points from previous domains $\cup_{j=1,...,i-1}\Omega_j$. While previous works on PINNs show that they often fail to learn the true solution due to loss imbalances (Wang et al., 2022a) or implicit biases towards higher times in collocation points during training (Wang et al., 2022a), our incremental strategy can alleviate these issues.

## 3.1 Theoretical analysis

Neural networks are universal approximators in the Sobolev space $W^{m,p}(\Omega)$ of $p$-integrable functions with $p$-integrable derivatives up to order $m$ (Hornik, 1991). The only requirement is for the activation function to be continuously differentiable $m$ times (as is the tanh function we will use). We now prove that learning the derivatives $D\boldsymbol{u}$, together with the IC and BC, is sufficient to learn $\boldsymbol{u}$. We show that minimizing the loss in equation 6 is equivalent to minimizing the distance between our solution $\hat{\boldsymbol{u}}$ and the true one $\boldsymbol{u}$. Our first result involves ODEs and one-dimensional functions, which resembles the fundamental theorem of calculus:

**Theorem 3.1.** *Let $u(t)$ be a (continuous) function in the space $W^{1,2}([0,T])$ on the interval $[0,T]$ and $\hat{u}(t)$ a neural network that approximates it. If the network is trained such that $L(u,\hat{u}) \to 0$, then $\hat{u} \to u$.*

We give the proof in Appendix B.1. The generalization of this result to higher dimensions turned out to be more challenging. We state the final result as:

**Theorem 3.2.** *Let $\Omega$ be a bounded open set and $u \in W^{1,2}(\Omega)$ a function in the Sobolev space with square norm. If the neural network $\hat{u}$ is trained such that $L(\hat{u},u) \to 0$, then $\|\hat{u}-u\|_{W^{1,2}(\Omega)} \to 0$ and $\|\hat{u}-u\|_{L^2(\partial\Omega)} \to 0$. To have a numerical approximation, if $L(\hat{u},u) \le \epsilon$, then $\|\hat{u}-u\|_{W^{1,2}(\Omega)} \le 2(C+1)\epsilon$ for some constant $C = C(\Omega)$. Furthermore, at the limit, the two functions coincide $\hat{u}_\infty = u$ inside $\Omega$ and at the boundary.*

The proof is in Appendix B.2. This result can be extended to include higher-order derivatives by adding the corresponding terms to the loss $L$, leading to a convergence in $W^{m,2}$, where $m$ is the highest order of derivative included. We use these theorems to prove that a neural network trained to optimize the loss in equation 6 learns the solution to a PDE. We prove the following theorem in Appendix B.3.

**Theorem 3.3.** *Let $\boldsymbol{u}$ be a solution to a PDE of the form of equation 3 with $\mathcal{F}$ continuous, and let $m$ be the order of the PDE, that is, the maximum order of derivative of $u$ that appears in $\mathcal{F}[\boldsymbol{u}]$. A Neural Network $\hat{\boldsymbol{u}}$ trained with the loss in equation 6, with derivatives terms up to order $m$, converges to the solution of the PDE $\boldsymbol{u}$ and fulfills all three conditions.*

*Remark* 3.4. PINNs incorporate the information on the PDE directly through the unsupervised PDE residual term $\|\mathcal{F}[\hat{\boldsymbol{u}}]\|_2^2$ in the loss. This way, PINNs have direct access to the definition of the physical system and the underlying equation representing its dynamics. On the contrary, DERL only learns the derivatives of the true solution $\boldsymbol{u}$ through the derivative learning term in equation 6, without seeing the true PDE. This theorem shows that learning the derivatives is enough for the DERL to recover a solution $\hat{\boldsymbol{u}}$ that fulfills the PDE, that is $\mathcal{F}[\hat{\boldsymbol{u}}] \to 0$. Hence, DERL learns to be consistent with the physical system without seeing the underlying equation. To provide an explicit example, consider a simple PDE such as $\mathcal{F}[u] = u_t + u_x = 0$ and let $u$ be a solution related to some specified initial and boundary conditions. As the loss in equation 6 converges to zero, we have that $\hat{u}_t \to u_t$ and $\hat{u}_x \to u_x$ as per Theorem 3.2. Hence, $\mathcal{F}[\hat{u}] = \hat{u}_t + \hat{u}_x \to u_t + u_x = \mathcal{F}[u] = 0$.

*Remark* 3.5. While perfect consistency requires enough derivative terms up to the order of the PDE, our results show that, in practice, the loss in equation 6 is enough for the majority of cases, already providing better results than the other baselines. This clearly shows that learning just $\boldsymbol{u}$, as in most supervised approaches, while derivatives already provide sufficient and richer information on the system.

When analytical derivatives are not available, we use empirical derivatives obtained via finite differences (Anderson et al., 2020): $\frac{\partial u}{\partial x_i}(\boldsymbol{x}) \simeq \frac{u(\boldsymbol{x}+he_i)-u(\boldsymbol{x})}{h}$, where $e_i$ is the unit vector in the direction of $x_i$ and $h$ is a

Table 1: Summary of the tasks we consider in the experiments.

| Experiment | Equation | Description | Tasks |
|---|---|---|---|
| **Allen-Cahn** | (AC)  $\lambda(u_{xx} + u_{yy}) + u(u^2 - 1) = f(\xi)$ | Second-order PDE | Generalize in-domain
Generalize to new parameters $\xi$
Transfer to a larger domain
Transfer to new parameters $\xi$ |
| **Continuity** | (CO)  $\frac{\partial u}{\partial t} + \nabla \cdot (\boldsymbol{v}u) = 0$ | Time-dependent PDE | Generalize in-domain |
| **Navier-Stokes** | (NS.M)  $[\mathrm{D}\boldsymbol{u}] \cdot \boldsymbol{u} - \Delta\boldsymbol{u} + \nabla p = 0$
(NS.I)$\nabla \cdot \boldsymbol{u} = 0$ | System of PDEs (Appendix) | Generalize in-domain |
| **Pendulum** | (DP)  $\ddot{u} + \frac{g}{l}\sin(u) + \frac{b}{m}\dot{u} = 0$ | ODE | Generalize to new
initial conditions $(u_0, \dot{u}_0)$ |
| **Korteweg-de Vries** | (KdV)  $u + uu_t + \nu u_{xxx} = 0$ | Third-order PDE | Transfer to a larger timespan
Higher-order distillation |

small positive real number. These approximations introduce errors, but the following result ensures that with a small enough $h$, these derivatives are very similar to the true ones. The proof is in Appendix B.4.

**Theorem 3.6.** *Let $u \in W^{1,2}(\mathbb{R}^n)$ and let $\mathrm{D}_{x_i}^\epsilon u$ be the difference quotient*

$$\mathrm{D}_{x_i}^\epsilon u(\boldsymbol{x}) = \frac{u(\boldsymbol{x} + h\boldsymbol{e}_i) - u(\boldsymbol{x})}{h} \tag{8}$$

*where $\boldsymbol{e}_i$ is the unit vector in the $x_i$ direction. Then, we have that $\|\mathrm{D}_{x_i}^\epsilon u\|_{L^2(\mathbb{R}^n)} \leq \|\frac{\partial u}{\partial x_i}\|_{L^2(\mathbb{R}^n)}$ and $\mathrm{D}_{x_i}^\epsilon u \to u_{x_i}$ in $L^2(\mathbb{R}^n)$, which means that empirical derivatives converge to weak (or true) ones as $\epsilon \to 0$. The same results hold for any open set $\Omega$, with the convergence being true on every compact subset of $\Omega$. In practical terms, empirical derivatives converge a.e. for every point distant at least $h$ from the boundary.*

*Remark* 3.7. The approximation used to calculate the derivative is of first order $O(h)$. If sufficient data is available, one can further improve precision by considering higher-order approximations, such as the central scheme (Anderson et al., 2020), which has an $O(h^2)$ error, or beyond.

## 4   Learning Physics from Data

We validate DERL on a set of dynamical systems and physical PDEs[1], investigating the generalization capabilities on a novel set of domain points (section 4.1) and on a novel set of parameters for parametric equations (section 4.2). During the training phase, the models did not have access to the set of points and parameters used to evaluate their generalization. Table 1 summarizes the systems we consider, with additional results in appendix C. For each experiment, we compare 4 approaches applied on a Multi-Layer Perceptron (MLP, Goodfellow et al. (2016)). Each of them learns the IC and the BC, together with a loss term for the domain $\Omega$ which is specific to the approach: (a) **DERL** (ours) optimizes equation 6, (b) **Output Learning (OUTL)** minimizes $\|u - \hat{u}\|_{L^2}$, (c) our extension of **Sobolev learning (SOB)** (Czarnecki et al., 2017) to dynamical systems include the loss term for both $u$ and $\mathrm{D}u$. For comparison, we also consider (d) OUTL+PINN, which consists of OUTL with an additional PINN loss term on the equation to improve its performance in the domain. The full description of the baselines and losses is in Appendix A. We evaluate all approaches with two metrics: (i) we measure the accuracy of the prediction by $\|u - \hat{u}\|_2$ - the $L^2$ distance between the true function $u$ and the estimate $\hat{u}$; (ii) considering MLPs as functions from the $\Omega$ to the domain of $u$, we measure how well a model satisfies the underlying physical equation $\mathcal{F}[u] = 0$. To this end, we compute the $L^2$ norm of the PDE residual of the network $\|\mathcal{F}[\hat{u}]\|_2$, as typically done with PINNs.

### 4.1   Generalization to new domain points

We test the ability of all approaches to learn the solution to a PDE in a domain $\Omega$. A fixed set of collocation points is used to train the models, which are then tested on an unseen grid of points at higher resolution.

---

[1]The code to reproduce the experiments is available at `https://github.com/AlexThirty/DERL`

Table 2: Results for the Allen-Cahn and continuity equation.

| Model | Task | $\|u - \hat{u}\|_2$ $\times 10^{-4}$ | $\|\mathcal{F}[\hat{u}]\|_2$ $\times 10^{-3}$ | Task | $\|u - \hat{u}\|_2$ $\times 10^{-2}$ | $\|\mathcal{F}[\hat{u}]\|_2$ $\times 10^{-1}$ |
|---|---|---|---|---|---|---|
| **DERL** (ours) | *Allen-Cahn* | **6.836** | **1.930** | *Continuity* | 9.059 | **2.247** |
| **OUTL** | | 56.95 | 11.41 | | **8.749** | 3.762 |
| **OUTL+PINN** | | 39.13 | 7.105 | | 9.378 | 2.8975 |
| **SOB** | | 9.448 | 2.469 | | 16.52 | 2.881 |

**Allen-Cahn Equation.** The Allen-Cahn equation (AC in table 1) is a non-linear PDE in the domain $[-1, 1]^2$. We consider $\lambda = 0.01$ and the analytical solution $u = \sin(\pi x)\sin(\pi y)$. The BC, the external force $f$, and the partial derivatives are calculated from $u$ using the PDE. The MLPs, which predict $u(x, y)$ from $(x, y)$, have 4 layers with 50 units (similar to Raissi et al. (2019)) and tanh activation and are trained for 100 epochs with the BFGS optimizer (Li & Fukushima, 2001) on 1000 random collocation points. Additional details are provided in Appendix C.1.

Table 2 shows the relevant metrics for this task. DERL outperforms all the other approaches in learning the true solution $u$, surpassing OUTL by a factor of 10. Furthermore, DERL achieves a lower PDE residual than OUTL+PINN. This shows that learning the direct solution $u$ is not as effective as DERL, not even when leveraging PINNs as an additional source of information. DERL also outperforms SOB, showing that derivatives contain all the needed information, while adding other terms to the loss can interfere and deteriorate the performance. This is also an empirical confirmation of our theoretical results. We finally report that DERL is the approach that best approximates the true derivatives ($\|\mathrm{D}u - \mathrm{D}\hat{u}\|_2$), surpassing OUTL and OUTL+PINN by a factor of 5 and improving on SOB. This is, of course, expected as DERL's loss target includes the derivatives. We provide additional results on this task in Appendix C.1.

**Continuity Equation.** We consider the continuity equation (equation CO in table 1), a time-dependent PDE describing mass conservation of matter in a velocity field. The domain of the unknown density $u(t, x, y)$ is the 2D plane region $[-1.5, 1.5]^2$ for $t \in [0, 10]$. The velocity field $\boldsymbol{v}$ is given by $\boldsymbol{v}(x, y) = (-y, x)$. The density is null at the boundary and the IC is given by 4 Gaussian densities (see Appendix C.2), as in Torres et al. (2024). The reference solution is calculated using the finite volumes method (Ferziger & Peric, 2001).The MLPs, which predict $u(t, x, y)$ from $(t, x, y)$, have the same architecture as the Allen-Cahn experiment and are trained for 200 epochs with batch size 128 and Adam optimizer (Kingma, 2014). Partial derivatives are *empirical* and were calculated by finite difference approximation. See C.2 for further details and results.

Table 2 reports the $L^2$ error on the solution $u$ and the PDE residual $L^2$ norm calculated on the finest grid available. DERL is the most effective method: the distance from the ground truth is second only to OUTL (although still comparable) and almost two times smaller than SOB. We conjecture this is due to its more complex optimization objective with possibly conflicting loss terms for $u$ and $\mathrm{D}u$. Plus, DERL is clearly best in terms of PDE residual, surpassing even OUTL+PINN, which means our model is more consistent with the underlying equation on domain points we have not trained the model on. On a side note, we report that the PINN loss alone fails to propagate the solution through time correctly, as reported in Appendix C.2. These results for PINNs are aligned with Wang et al. (2022a), where the authors conjectured that PINN's gradients are biased towards high values of $t$ with the model failing to propagate information correctly from the IC.

**Navier-Stokes.** We now consider a well-known system of PDEs: 2D time-independent Navier-Stokes equations made of the momentum (NS.M) and incompressibility (NS.I) equations in Table 1. The domain is $(x, y) \in [-1, 1] \times [-0.5, 1.5]$ with the viscosity coefficient set to $\nu = \frac{1}{50}$. The particular solution we work with is the so-called Kovasznay flow (Drazin & Riley, 2009), representing the flow around a two-dimensional grid, with the analytical form given in appendix C.3. Training data is sampled from a regular grid of points with $dx = dy = 0.01$, while testing data comes from a finer one, with $dx = dy = 0.005$. We work with MLPs with 4 layers of 64 units and tanh activation, trained with the LBFGS optimizer for 100 epochs. Here, we also measure the error on vorticity $\omega = \nabla \times \boldsymbol{u} = \sin x \sin y e^{-2\nu t}\hat{\boldsymbol{z}}$, which is a vector that describes the rotating motion of a fluid at a given point and is of particular interest for turbulent flows such as this experiment.

Table 3: Results for the Navier-Stokes Kovasznay flow experiment.

| Model | $\|(\boldsymbol{u},p) - (\hat{\boldsymbol{u}},\hat{p})\|_2$ $\times 10^{-5}$ | $\|\omega - \hat{\omega}\|_2$ $\times 10^{-4}$ | **(NS.M)** $\|\mathcal{F}[\hat{\boldsymbol{u}},\hat{p}]\|_2$ $\times 10^{-4}$ | **(NS.I)** $\|\mathcal{F}[\hat{\boldsymbol{u}}]\|_2$ $\times 10^{-4}$ |
|---|---|---|---|---|
| **DERL** (ours) | **0.6719** | **0.6611** | 0.8810 | 0.5945 |
| **OUTL** | 9.201 | 106.6 | 10.58 | 10.30 |
| **OUTL+PINN** | 3.331 | 5.042 | **0.8465** | **0.5729** |
| **SOB** | 1.125 | 1.077 | 1.660 | 1.177 |

Figure 3: $L^2$ error difference in the learned solution $(\hat{\boldsymbol{u}}, \hat{p})$ between each methodology and DERL. The blue area is where DERL performs better than the comparison.

Numerical results are available in Table 3. DERL is the best overall model, achieving the lowest $L^2$ errors on both the solution $\boldsymbol{u}, p$ and the vorticity $\omega$. DERL also follows closely the PDE residuals of OUTL+PINN, while OUTL alone fails to reproduce the vorticity of the solution. Figure 11 shows the difference between the method's error and the one by DERL, with the blue color meaning DERL is performing better in the region. This figure clearly shows that learning only $\boldsymbol{u}$ does not provide enough information on the physical systems, especially in this turbulent case.

## 4.2 Generalization to new ODE/PDE parameters

We now shift towards more complex tasks involving solutions to parametric ODEs and PDEs of the form $\mathcal{F}(\boldsymbol{u}; \xi) = 0$, where $\xi \in \mathbb{R}^l$ is the set of parameters. The objective of the models is to generalize to a set of parameters unseen during training. A supervised machine learning pipeline is performed where the set of parameters (and their solutions) is divided into training, validation, and testing. We tuned the models using the training and validation sets, with the final results given by the testing set. All models take as input the spatiotemporal coordinates and the parameters $(t, \boldsymbol{x}, \xi)$.

**Damped Pendulum.** We consider the dynamical system of a damped pendulum with state equation (DP) in table 1. Additional details on the experimental setup are available in Section C.4. We sampled a total of 50 trajectories from different starting conditions $(u_0, \dot{u}_0)$, which act as parameters: 30 are reserved for training, 10 for validation, and 10 for testing. Each trajectory was sampled every $\Delta t = 0.01s$. The MLPs, which predict $u(t; u_0, \dot{u}_0)$ from $(t, u_0, \dot{u}_0)$, have 4 MLP with 50 units each and tanh activation, and are trained for 200 epochs with batch size 64. Initial states are sampled randomly in $(u_0, \dot{u}_0) \in [-\frac{\pi}{2}, \frac{\pi}{2}] \times [-1.5, 1.5]$. In addition to the previously-used metrics and the angular velocities $\|\dot{u} - \hat{\dot{u}}\|_2$, we also monitor:
∘ $\|\mathcal{G}[\hat{u}]\|_2$ (formally defined in Appendix C.4.1), the PDE residual which describes continuity and differentiability of trajectories with respect to the initial conditions. High values of $\|\mathcal{G}[\hat{u}]\|_2$ imply predicted trajectories do not change smoothly when varying $(u_0, \dot{u}_0)$, which is not consistent with the real system.
∘ The error in the learned field at $t = 0$, that is, the distance between true and predicted derivatives $\|\mathcal{F}[\hat{u}]_{t=0}^{\text{full}}\|_2$ for every possible initial state in the domain. This measures how well the models predict the dynamics for different initial conditions.

We report numerical results in Table 4. Figure 4a shows a direct comparison of the errors in the learned field $\mathcal{F}[\hat{u}_0]$ at $t = 0$ in the whole domain. We computed the local error on grid points for each method. Then, for each method, we computed the difference between its errors and DERL's error, such that positive values (blue) are where DERL performs better. DERL is second best and very close to OUTL at predicting the

Table 4: Numerical results for the damped pendulum.

| Model | $\|u - \hat{u}\|_2$ $\times 10^{-2}$ | $\|\dot{u} - \hat{u}\|_2$ $\times 10^{-2}$ | $\|\mathcal{F}[\hat{u}]\|_2$ $\times 10^{-2}$ | $\|\mathcal{G}[\hat{u}]\|_2$ $\times 10^{-2}$ | $\|\mathcal{F}[\hat{u}]_{t=0}^{\text{full}}\|_2$ $\times 10^{-1}$ |
|---|---|---|---|---|---|
| **DERL** (ours) | 1.069 | **1.296** | **1.277** | **9.244** | **4.445** |
| **OUTL** | **0.9607** | 5.766 | 5.452 | 20.55 | 8.233 |
| **OUTL+PINN** | 1.118 | 7.614 | 7.070 | 30.07 | 10.65 |
| **SOB** | 1.133 | 2.824 | 2.803 | 15.17 | 8.054 |

(a)                                              (b)

Figure 4: Pendulum experiment. $L^2$ error difference in the learned field at $t = 0$ between each methodology and DERL. The blue area is where DERL performs better than the comparison.

trajectories and outperforms by a large margin all other approaches in predicting the angular velocity of the pendulum. The trajectories predicted by DERL have also the lowest $\mathcal{F}[\hat{u}]$ and $\mathcal{G}[\hat{u}]$ norms. This shows that among all models, DERL's predictions are the most physical ones and the most consistent with the underlying equations. Figure 4a shows that DERL learns more accurately the field for every initial condition. For this task, we adapted Lagrangian Neural Networks (LNN) (Cranmer et al., 2019) and Hamiltonian Neural Networks (HNN) (Greydanus et al., 2019) to the damped pendulum case. We train them on the conservative part of the field (see C.4.2 for details), where they excel as they are specifically designed for conservative fields. Unlike DERL, they also require trajectories to be calculated through external solvers. Remarkably, DERL outperforms both LNN and HNN (figure 4b) on the learned field: DERL scores an error of 0.4445, against 0.4470 and 0.4496 for HNN and LNN, respectively.

**Single-parameter Allen-Cahn Equation.** Inspired by Penwarden et al. (2023), we consider a parametric experiment on the Allen-Cahn equation (AC). The parametric component is the external forcing $f$, calculated from the analytical solution $u(x, y; \xi) = e^{-\xi(x+0.7)} \sin(\pi x) \sin(\pi y)$, with parameter $\xi \in \mathbb{R}$. We reserved 8 values of $\xi$ for training, 2 for validation, and 5 for testing, and we used 1000 collocation points in the domain. The architecture is the same as in the Allen-Cahn experiment in Section 4.1. For further details and the complete setup, see Appendix C.5. Numerical results on the testing parameters are reported in Table 5, while further results are available in Appendix C.5.1. DERL always scores the best results, highlighting strong generalization abilities to unseen parameterization of the PDE. OUTL underperforms in both metrics, showcasing that learning from the raw data $u$ often leads to misleading solutions that do not follow the underlying physical laws. SOB underperforms as it again struggles to optimize a loss with both $u$ and $\mathrm{D}u$.

**Double-parameter Allen-Cahn equation.** We now increase complexity by considering a dual-parameter version of the Allen-Cahn equation (Zou & Karniadakis, 2023). The analytical solution is given by $u(x, y; \xi_1, \xi_2) = \sum_{j=1}^2 \xi_j \frac{\sin(j\pi x)\sin(j\pi y)}{j^2}$, from which the external forcing $f$ is calculated. We selected 15 random couples of the parameters $(\xi_1, \xi_2) \in [0,1]^2$ for training, 5 for validation and 5 for test. Numerical results for the test parameters are reported in Table 5. Once again, DERL is consistently the best model for every couple of parameters (see Appendix C.5.2), performing at least two times better in predicting $u$ for unseen ones. Learning $u$ through OUTL leads to very poor performance in terms of physical consistency, even with the additional help of the PINN loss.

Table 5: Results for the parametric Allen-Cahn experiments, averaged over test parameters.

| Model | Single-Parameter | | Double-parameter | |
|---|---|---|---|---|
| | $\|u - \hat{u}\|_2$ $\times 10^{-3}$ | $\|\mathcal{F}[\hat{u}]\|_2$ $\times 10^{-2}$ | $\|u - \hat{u}\|_2$ $\times 10^{-3}$ | $\|\mathcal{F}[\hat{u}]\|_2$ $\times 10^{-2}$ |
| **DERL** (ours) | **5.441** | **1.730** | **9.628** | **1.467** |
| **OUTL** | 8.486 | 11.67 | 19.13 | 8.050 |
| **OUTL+PINN** | 6.413 | 2.466 | 27.70 | 8.446 |
| **SOB** | 9.736 | 2.276 | 21.26 | 1.477 |

## 5 Transferring physical knowledge

We have shown that DERL is capable of extracting the relevant physical features from data.We now show how we can take advantage of this to transfer physical information between models through distillation (Hinton et al., 2015), eventually building physical models incrementally.
We set up our transfer experiments by dividing the whole training experience into 2 or 3 *incremental steps*. Each step extends the physical problem to i) new portions of the time domain, ii) to new portions of the space domain, and iii) to new parameterizations of the PDE. For these incremental training experiments, we assume an unsupervised setting where ground truth data is not available, and the models have to find a solution only by the definition of the PDE problem, as PINNs do.

The physical models that we train and use as teachers are all PINNs. For the distillation component in equation 7, we tested the DERL, SOB, and OUTL losses. During the first step, we train a model from scratch on the given problem. From the second step on, we consider the *teacher* the model trained on the previous step, and we transfer its knowledge to a student model. The student is either a randomly initialized new model (*from-scratch* in tables) or the previously trained model (*continual*), being fine-tuned. We use our distillation approach to preserve the teacher's knowledge while the student learns the current step.
As a comparison, we train three additional PINN models. PINN-no-distillation updates the teacher with the PINN loss on the new domain, but without distillation on the previous domain, and is expected to forget previous information. PINN-full is a PINN trained on the *full* domain $\Omega$ without the iterative distillation process. This model is trained for the total sum of epochs of the other methodologies for a fair comparison. In continual learning, this joint training method is often taken as an upper bound for the expected performance of the final continual learning model Masana et al. (2023). Finally, we consider PINN-replay, which updates the teacher with the PINN loss on the new domain and on a subset of points (20%) from the previous domain. This is our closest resemblance to an Experience Replay setting.

### 5.1 Transfer to wider time horizons

In this first experiment with transferring, we aim to extend the working domain of a PINN in time. We consider the 1D Korteweg-de Vries equation (KdV) in Table 1. The spatial domain is the interval $[-1, 1]$. while the entire time domain is $[0, 1]$. We first train a PINN in the time interval $[0, 0.35]$ and we collect its evaluations $\hat{u}, D\hat{u}$ on the training collocation points in a dataset. Then, a second model is trained with the combination of the PINN loss on the second time interval $[0.35, 0.7]$, and a distillation loss on the old collocation points. This procedure aims to transfer old knowledge to this new model, while discovering the solution in a new interval of the time domain. Hence, this model should be able to learn the solution in the time-space domain $(t, x) \in [0, 0.7] \times [-1, 1]$. Finally, we repeat this procedure one more time, extending the model to the time interval $[0.7, 1]$. As we will see, the final model has correctly learned the solution in the entire time-space domain of $[0, 1] \times [-1, 1]$. For this task, each model is an MLP with 9 layers of 50 units and tanh activation, due to the more complex nature of the KdV task. Training phases last 30.000, 30.000, and 40.000 steps each with the Adam optimizer and exponentially decaying lr, starting from 0.001 with factor 0.9. The PINN-full, PINN-no-distillation, and PINN-replay models train for the sum of these steps, that is 100.000, with the same optimizer and learning rate schedule.

Table 6: Results for the physical knowledge transfer experiments. $L^2$ error and PDE residual on the whole domain (or on test parameters) at the final stage of incremental training.

| **Model** | | Larger timespan 5.1 | | Larger domain 5.2 | | New parameters 5.3 | |
| | | $\|u - \hat{u}\|_2$ $\times 10^{-2}$ | $\|\mathcal{F}[\hat{u}]\|_2$ $\times 10^{-1}$ | $\|u - \hat{u}\|_2$ $\times 10^{-3}$ | $\|\mathcal{F}[\hat{u}]\|_2$ $\times 10^{-3}$ | $\|u - \hat{u}\|_2$ $\times 10^{-3}$ | $\|\mathcal{F}[\hat{u}]\|_2$ $\times 10^{-3}$ |
|---|---|---|---|---|---|---|---|
| **DERL** | from-scratch | **3.331** | **2.932** | **3.332** | **7.453** | **7.736** | **7.537** |
| | continual | **2.379** | 1.731 | **2.377** | **3.873** | 11.76 | **7.275** |
| **OUTL** | from-scratch | 9.302 | 8.966 | 6.850 | 15.10 | 9.757 | 10.27 |
| | continual | 9.334 | 7.571 | 3.399 | 8.995 | 11.49 | 11.87 |
| **SOB** | from-scratch | 3.662 | 4.327 | 5.151 | 9.480 | 11.03 | 11.63 |
| | continual | 2.535 | **1.536** | 2.642 | 5.284 | **8.840** | 7.645 |
| **PINN** | full | 9.801 | 1.730 | 2.356 | 6.275 | 14.96 | 6.302 |
| **PINN** | no distillation | 15.19 | 18.29 | 247.3 | 245.6 | 31.87 | 16.14 |
| **PINN** | replay | 5.371 | 1.889 | 5.281 | 10.54 | 13.72 | 6.289 |

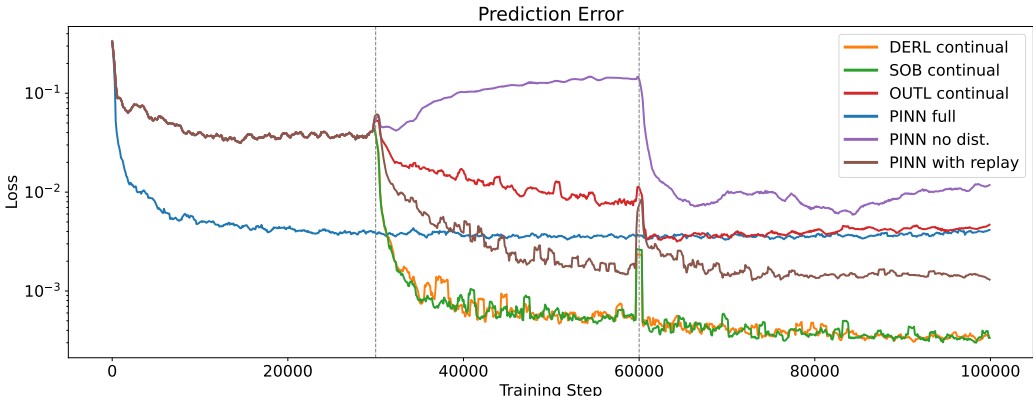

Figure 5: Prediction errors for the KdV transfer experiment for the continual models and the PINN variants. Vertical lines represent when a new step of incremental training is initiated.

Numerical results are available in Table 6. DERL outperforms all other methodologies in learning the solution to the PDE, either starting from-scratch or by fine-tuning the teacher model. Most importantly, our incremental methodology, coupled with the DERL distillation loss, improves the results by three times compared to a standard PINN. Figure 5 shows the prediction error in the *whole* domain as a function of the cumulative training steps for the *continual* models. A similar plot including the *from-scratch* models is available in Appendix D.3. These losses follow the typical behavior of continual learning, where the loss increases at the beginning of each phase and subsequently decreases, with the difference that performance is measured on the entire final task. The results clearly show that the PINN reaches a minimum at the beginning of training and cannot improve, while our incremental approach performs almost an order of magnitude better in the end. While PINN-replay improves on the baseline, using DERL to distill the teacher derivatives performs better to retain knowledge compared to experience replay. We notice that, even for transferring, it is not enough to distill the model's outputs, both in terms of consistency and prediction error. Derivatives are both necessary and sufficient to distill the physical knowledge correctly. Finally, we see that incrementally training a PINN without distilling previous knowledge leads to forgetting of the learned solution. See Appendix D.3 for additional figures.

## 5.2 Transfer to new domains

We leverage our transfer protocol to extend the working domain of a model (Figure 1). We consider the Allen-Cahn equation (AC) from Section 4.1. We first train a PINN in the lower-left quadrant of the domain

$([-1,0] \times [-1,0])$. We evaluate $\hat{u}, D\hat{u}$ on the training collocation points. Then, a second model is trained with a combination of the PINN loss on the lower-right quadrant $([0,1] \times [-1,0])$, and a distillation loss for the old points. The resulting model should be able to learn the solution in the lower half of the domain $([-1,0] \times [1,0])$. The same procedure is repeated to extend the working domain of the model to the upper half $([-1,1] \times [0,1])$. The final model should be a good predictor in the entire domain $([-1,1] \times [-1,1])$. We used the same architecture as in Section 4.1 and 100 epochs per step with the BFGS optimizer (Appendix D.4).

The numerical results in table 6 show that DERL enables the best-performing distillation, achieving the lowest error on $u$ and the best PDE residual in both the *from-scratch* and *continual* case. DERL even beats the PINN trained on the whole domain for the same number of epochs, especially in terms of PDE residual. This is surprising, as the PINN trained on the whole domain would tipically be used as an upper bound in continual learning (the so-called "multitask" model) (Masana et al., 2023). Instead, DERL surpasses its performance, showcasing promising continual learning abilities.

### 5.3 Transfer to new PDE parameters

We consider the Allen-Cahn equation (AC) with two parameters $(\xi_1, \xi_2)$ as in Section 4.2. We first trained a PINN on 1000 random collocation points for 10 couples of $(\xi_1, \xi_2)$. Then, on the remaining 10 couples we trained a new model, either from scratch or by fine-tuning the previous one. We train with a PINN loss on the new parameters and we use distillation with DERL, OUTL or SOB.

Table 6 reports the $L^2$ errors and PDE residuals on the test parameters for both the *from-scratch* and *continual* cases. DERL proves to be the best distillation protocol. It also surpasses all PINN models, except for the full and replay PINNs on the PDE residual metric, whilst being the closest to match its performance. This again shows our method's potential to transfer physical information across different instances of PDEs.

### 5.4 Distillation of higher-order derivatives

Finally, we investigate the effects of distilling higher-order derivatives. We consider the Korteweg-de Vries (KdV) equation (KdV in table 1), a third-order non-linear PDE. A pre-trained PINN teacher distills knowledge to students with identical architecture. We tested Hessian learning (HESL), which distills second-order derivatives and its combination with DERL (DERL+HESL) and SOB (SOB+HESL). The full setup and results can be found in section D.3. Here, we summarize the main findings. We observe that OUTL *fails* to distill effectively, as its PDE residual of 17.366 is two orders of magnitude larger than the other models. Interestingly, the best-performing methods, HESL and DERL+HESL, did not access $u$ directly. They achieve the lowest PDE residual (0.1915 and 0.1931, respectively) which is comparable to the PINN PDE residual of 0.1664. Both approaches score the same error on $u$ as the teacher model, and even a lower BC loss.

## 6 Conclusion

We proposed DERL, a supervised methodology to learn physical systems from their partial derivatives. We showed theoretically and experimentally that our method successfully learns the solution to a problem and remains consistent with its physical constraints, outperforming other supervised learning methods. We also found DERL to be effective in transferring physical knowledge across models through distillation. We leveraged this capability to extend the working domain of a PINN and to generalize better across different PDE parameterizations. Finally, we showed the advantage of distilling higher-order derivatives. Our work shows promising intersections with continual learning, as DERL's distillation allowed us to build physical models incrementally, showing minimal to no forgetting of previous knowledge. In the future, learning physical and dynamical systems may not always require starting from scratch. Rather, the learning process may be based on the continual composition and integration of physical information across different expert models. A more general and flexible paradigm that could lead to data-efficient and sustainable deep learning solutions.

**Acknowledgments**

This work has been supported by EU-EIC EMERGE (Grant No. 101070918).

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

## A  Implementation Details and Setups

In this Section, we give additional details on the implementation of the models and their respective losses. We start by defining the individual loss components in the most general case. Each experiment will feature its required terms based on the problem definition.

**Sobolev norms.**  Let $u(\boldsymbol{x}) : \Omega \to \mathbb{R}$ be the function we want to learn. In the case of $u$ taking values in $\mathbb{R}^D$, we can take the sum over the vector components. The main loss terms are those linked to learning the solution in the domain $\Omega$. The distance in the Sobolev space $W^{m,2}(\Omega)$ between the function $u$ and our Neural Network $\hat{u}$ is defined as

$$\|u - \hat{u}\|^2_{W^{m,2}(\Omega)} = \|u - \hat{u}\|^2_{L^2(\Omega)} + \sum_{l=1}^{m} \left\| \mathrm{D}^l u - \mathrm{D}^l \hat{u} \right\|^2_{L^2(\Omega)}, \tag{9}$$

where $\mathrm{D}^l u$ is the $l$-th order differential of $u$, i.e. the gradient or Jacobian for $l = 1$ and the Hessian for $l = 2$. Practically, these squared norms are approximated via Mean Squared Error (MSE) on a dataset $\mathcal{D}_d$ of collocation points $\boldsymbol{x}_i \in \Omega, i = 1, \ldots, N_d$ with the respective values of the function and its differentials:

$$\mathcal{D}_d = \left\{ (\boldsymbol{x}_i, u(\boldsymbol{x}_i), \mathrm{D}u(\boldsymbol{x}_i), \ldots, \mathrm{D}^l u(\boldsymbol{x}_i)) \right\}, \quad \boldsymbol{x}_i \in \Omega, \quad i = 1, \ldots, N_d. \tag{10}$$

The MSE is then calculated as

$$\|u - \hat{u}\|^2_{L^2(\Omega)} \simeq \frac{1}{N_d} \sum_{i=1}^{N_d} (u(\boldsymbol{x}_i) - \hat{u}(\boldsymbol{x}_i))^2$$

$$\left\| \mathrm{D}^l u - \mathrm{D}^l \hat{u} \right\|^2_{L^2(\Omega)} \simeq \frac{1}{N_d} \sum_{i=1}^{N_d} \left( \mathrm{D}^l u(\boldsymbol{x}_i) - \mathrm{D}^l \hat{u}(\boldsymbol{x}_i) \right)^2, \quad l = 1, \ldots, m \tag{11}$$

Each of the models will feature one or more terms in equation 11. In particular, OUTL uses only the first, DERL, HESL and our other methodologies use only the second, with one or more values of $l \neq 0$, and SOB uses both. In the case of time-dependent problems, it is sufficient to consider the augmented domain $\tilde{\Omega} = [0, T] \times \Omega$ and collocation points that are time-domain couples $(t_i, \boldsymbol{x}_i), i = 1, \ldots, N_d$. The models' derivatives are calculated using Automatic Differentiation (Baydin et al., 2018) with the python PyTorch package (Paszke et al., 2019) and, in particular, its functions `torch.func.jacrev` and `torch.func.hessian`. In the case of SOB, the derivative terms of order $l \geq 2$ are approximated using discrete difference expectation over random vectors (see Czarnecki et al. (2017) and Rifai et al. (2011) for more details).

**PDE residuals.**  For the evaluation of the physical consistency of a model and OUTL+PINN, we adopt the $L^2$ norm of the MLPs' PDE residual, as in Raissi et al. (2019). Let the problem be defined by a Partial Differential Equation $\mathcal{F}[u(\boldsymbol{x})] = 0$, where $\mathcal{F}$ is a differential operator

$$\mathcal{F}[u(\boldsymbol{x})] = f(\boldsymbol{x}) + a_0(\boldsymbol{x}) u(\boldsymbol{x}) + \sum_{i=1}^{n} a_i(\boldsymbol{x}) \frac{\partial u}{\partial x_i}(\boldsymbol{x}) + \sum_{i=1}^{n} \sum_{j=1}^{n} a_{ij}(\boldsymbol{x}) \frac{\partial^2 u}{\partial x_i \, \partial x_j}(\boldsymbol{x}) + \ldots \tag{12}$$

and $f(\boldsymbol{x})$ is a function independent of $u$, usually called (external) *forcing*. The PDE residual measure we consider is the $L^2$ norm of $\mathcal{F}[\hat{u}]$, which is approximated again as the MSE over collocation points dataset $\mathcal{D}_d$ defined above. In this case, there is no supervised target and the operator is applied to the model using AD, obtaining the loss term

$$\|\mathcal{F}[\hat{u}]\|^2_{L^2(\Omega)} = \frac{1}{N_d} \sum_{i=1}^{N_d} (\mathcal{F}[\hat{u}(\boldsymbol{x}_i)])^2. \tag{13}$$

This loss will be used to measure the consistency of our model to the physics of the problem or the PDE in general in a strong sense since it contains the derivatives of the network itself.

**Boundary (and Initial) conditions.** To learn the function $u(\boldsymbol{x})$, as per theorem 3.2, we need to provide some information at the boundary $\partial\Omega$ as well. Otherwise, we can only conclude that $u(\boldsymbol{x})$ and $\hat{u}(\boldsymbol{x})$ differ by a possibly non-zero constant. In one dimension, this reminds us of the indefinite integral $F(x) = \int f(x) + C$, which is defined up to a constant $C$ determined by some condition on the function integral $F$. To measure the $L^2$ distance between the true function and our approximation on the boundary, we employ the usual $L^2(\partial\Omega)$ norm approximated via MSE loss on a dataset of collocation points

$$\mathcal{D}_b = \left\{ (\boldsymbol{x}_i, u(\boldsymbol{x}_i) = b(\boldsymbol{x}_i)) \right\}, \qquad \boldsymbol{x}_i \in \partial\Omega, \qquad i = 1, \ldots, N_b, \tag{14}$$

where $b(\boldsymbol{x})$ is a function defining the boundary conditions. The norm is then approximated as

$$\|u - \hat{u}\|_{L^2(\partial\Omega)} \simeq \frac{1}{N_d} \sum_{i=1}^{N_b} \left( b(\boldsymbol{x}_i) - \hat{u}(\boldsymbol{x}_i) \right)^2. \tag{15}$$

In the case of time-dependent problems, we consider the augmented domain $\tilde{\Omega} = [0, T] \times \Omega$. Here, the collocation points of the boundary conditions are time-space couples $(t_i, \boldsymbol{x}_i) \in [0, T] \times \partial\Omega$. For time-dependent problems, initial conditions $u(0, \boldsymbol{x}) = g(\boldsymbol{x})$ are also required as part of the extended boundary and are learned with

$$\|u(0, \boldsymbol{x}) - \hat{u}(0, \boldsymbol{x})\|_{L^2(\Omega)} \simeq \frac{1}{N_i} \sum_{i=1}^{N_i} \left( g(\boldsymbol{x}_i) - \hat{u}(0, \boldsymbol{x}_i) \right)^2. \tag{16}$$

In theory, the augmented boundary should contain the values of $u(T, \boldsymbol{x})$ at the final time $T$. However, PDE problems are usually defined without a final condition, and it is more interesting not to have one to see which methodology can propagate physical information better. For example, SOTA methods as PINN do not include them (Raissi et al., 2019). Therefore, we decided not to include them as well, leading to a harder experimental setup than our theoretical results. For example, the experimental results in Appendix C.2 where the PINN failed to propagate the solution from $t = 0$ to higher times, confirm the findings of Wang et al. (2022a) of PINNs being biased during training. However, DERL did not suffer from this issue.

Each of the above components, when present, will have its weight in the total loss, which is one hyperparameter to be tuned. Below we report details on the baseline models and their losses.

**Details for the baseline models.** The baselines used in the main text comprehend:

- OUTL, that is supervised learning of the solution $u$.

- OUTL+PINN, a combination of supervised learning of $u$ and the additional information provided by the PINN loss (Raissi et al., 2019). This provides the model with information on the underlying PDE and should increase its physical consistency.

- SOB, that is Sobolev learning, first introduced in Czarnecki et al. (2017), which considers the both the losses on $u$ and its derivatives, as in equation 11.

**Loss comparison.** We report here the losses used for each methodology to have a clean comparison between them. For space reasons, we write the true norms instead of the MSE approximations described above. Each norm is in $L^2(\Omega)$ or $L^2([0, T] \times \Omega)$ based on the setting.

$$\begin{aligned}
OUTL : &L(u, \hat{u}) = \lambda_D \|u - \hat{u}\|^2 + \lambda_I \text{IC} + \lambda_B \text{BC} \\
OUTL + PINN : &L(u, \hat{u}) = \lambda_D \|u - \hat{u}\|^2 + \lambda_P \|\mathcal{F}[\hat{u}]\|^2 + \lambda_I \text{IC} + \lambda_B \text{BC} \\
DERL : &L(u, \hat{u}) = \lambda_D \|\mathrm{D}u - \mathrm{D}\hat{u}\|^2 + \lambda_I \text{IC} + \lambda_B \text{BC} \\
DER + HESL : &L(u, \hat{u}) = \lambda_D \left( \|\mathrm{D}u - \mathrm{D}\hat{u}\|^2 + \|\mathrm{D}^2 u - \mathrm{D}^2 \hat{u}\|^2 \right) + \lambda_I \text{IC} + \lambda_B \text{BC} \\
HESL : &L(u, \hat{u}) = \lambda_D \|\mathrm{D}^2 u - \mathrm{D}^2 \hat{u}\|^2 + \lambda_I \text{IC} + \lambda_B \text{BC} \\
SOB : &L(u, \hat{u}) = \lambda_D \left( \|u - \hat{u}\|^2 + |\mathrm{D}u - \mathrm{D}\hat{u}\|^2 \right) + \lambda_I \text{IC} + \lambda_B \text{BC} \\
SOB + HES : &L(u, \hat{u}) = \lambda_D \left( \|u - \hat{u}\|^2 + \|\mathrm{D}u - \mathrm{D}\hat{u}\|^2 + \|\mathrm{D}^2 u - \mathrm{D}^2 \hat{u}\|^2 \right) + \lambda_I \text{IC} + \lambda_B \text{BC} \\
PINN : &L(u, \hat{u}) = \lambda_D \|\mathcal{F}[\hat{u}]\|^2 + \lambda_I \text{IC} + \lambda_B \text{BC}
\end{aligned} \tag{17}$$

Table 7: Computational time of 1 epoch for each experiment with the Adam optimizer. Allen-Cahn results are for the experimental setup in Appendix C.1. For NCL we reported the computational time of 100 steps due to the definition and length of one epoch as in Richter-Powell et al. (2022).

| Model | Pendulum | Allen-Cahn | Continuity | KdV | NCL |
|---|---|---|---|---|---|
| **OUTL+PINN** | 6.136 | 4.009 | 12.61 | 37.57 | / |
| **DERL** | 3.599 | 2.575 | 5.867 | 17.85 | 8.358 |
| **HESL** | / | / | / | 25.43 | / |
| **DER+HESL** | / | / | / | 28.75 | / |
| **OUTL** | 3.376 | 2.442 | 5.380 | 14.73 | 7.826 |
| **SOB** | 3.331 | 2.564 | 5.887 | 18.17 | 8.3365 |
| **SOB+HES** | / | / | / | 24.34 | / |

where BC and IC are respectively given by equation 15 and equation 16.

## A.1 Computational time

Although effective, optimization of higher-order derivatives of a Neural Network can be costly in time and computational terms. On the other hand, the functional and the Automatic Differentiation framework of Pytorch (Paszke et al., 2019) allowed us to calculate such quantities with ease and in contained time. For a complete comparison, we calculated the time to perform one training epoch with the Adam optimizer for each task and model. Results are available in table 7. As expected, OUTL is the fastest having the most basic objective, while DERL beats both SOB and OUTL+PINN. It seems clear that the more terms in equation 17, the longer the training time, as seen by adding the Hessian matrices in the KdV experiment. Here, the third-order derivatives have a huge impact on PINN training. On the other hand, the approximations employed in SOB to calculate the Hessian matrices did not significantly improve the computation time compared to our approach with Automatic Differentiation. For reference, all the experiments were performed on an NVIDIA H100 GPU.

## A.2 Model tuning

For the OOD generalization experiments in Section 4.2, that is the two parametric versions of Allen-Cahn and the pendulum experiments, each model was tuned individually on the task to find the best hyperparameters. To do so, we performed a standard train/validation/test pipeline. This was done to extract the best performance, to have a fair comparison among the methodologies, and to measure their effectiveness in the tasks. The hyperparameters to be tuned are given by the weights of the different norms in equation 17, as well as the learning rate. The batch size was fixed for each experiment beforehand.

The tuning was conducted using the Ray library (Moritz et al., 2018) with the HyperOpt search algorithm (Bergstra et al., 2013) and the ASHA scheduler (Li et al., 2020a) for early stopping of unpromising samples. For each weight $\lambda_D, \lambda_I, \lambda_B$, the search space was the interval $[10^{-3}, 10^2]$ with log-uniform distribution, and for the learning rate, we consider discrete values in $[0.0001, 0.0005, 0.001]$. Most importantly, the target metric of the tuning to evaluate individual runs was the $L^2$ loss on the function $\|u - \hat{u}\|_{L^2(\Omega)}$, being the usual target in data-driven tasks.

For the pendulum experiments that involve full trajectories, 30 trajectories are for training and 10 for validation, while 10 unseen trajectories are used for model testing. For the pendulum interpolation task in Section C.4, a total of 40 trajectories are used for in the tuning phase: points with times between $t = 3.75$ and $t = 6.25$ are unseen by the models and used for tuning/validation, while 10 unseen complete trajectories are used for testing. For the parametric Allen-Cahn experiment with a single parameter, we calculated the solution on a set of 1000 collocation points for 13 values of $\xi \in [0, \pi]$: 8 are used for training, 2 for validation and 5 for testing. Similarly, for the experiment with two parameters, we sampled 25 couples of $(\xi_1, \xi_2) \in [0, 1]^2$ with a 60/20/20 split.

Finally, a tuning procedure was applied to find the optimal weights for the experiments in Section 4.1 trained with the Adam optimizer. In these cases, no information on the test set is used and weights are found by

optimizing the error solely on training points, as it is in common practice for physical systems. This ensures there is no information leakage from the test set.

We remark that no tuning is performed for the experiments with distillation and physical knowledge transfer.

# B  Proofs of the Theoretical Statements

## B.1  Proof of theorem 3.1

*Proof.* Since $u(t) \in W^{1,2}([0,T])$, $u$ is actually Holder continuous or, to be precise, it has a Holder continuous representative in the space (Maz'ya, 2011). Since $L(u, \hat{u}) \to 0$, we have that $\hat{u}'(t) \xrightarrow{L^2([0,T])} u'(t)$ and $\hat{u}(0) \to u(0)$ (the initial conditions are just one point). If $v = u - \hat{u}$, we have that $v' \xrightarrow{L^2} 0$ and $v$ converges to a function a.e. constant, which we can suppose to be continuous as above. Then, since $v(0) = 0$, we have that $v \equiv 0$ and $\hat{u} \equiv u$. $\square$

## B.2  Proof of theorem 3.2

We begin by stating Poincaré inequality (see also Evans (2022) and Maz'ya (2011)).

**Theorem B.1** (Evans (2022) Section 5.6, theorem 3 and Section 5.8.1, theorem 1). *Let $1 \le p < \infty$ and $\Omega$ be a subset bounded in at least one direction. Then, there exists a constant $C$, depending only on $\Omega$ and $p$, such that for every function $u$ of the Sobolev space $W_0^{1,p}(\Omega)$ of functions null at the boundary, it holds:*

$$\|u\|_{L^p(\Omega)} \le C\|\mathrm{D}u\|_{L^p(\Omega)} \tag{18}$$

*In case the function is not necessarily null at the boundary $u \in W^{1,p}(\Omega)$ and $\Omega$ is bounded, we have that*

$$\|u - (u)_\Omega\|_{L^p(\Omega)} \le C\|\mathrm{D}u\|_{L^p(\Omega)} \tag{19}$$

*where $(u)_\Omega = \frac{1}{|\Omega|} \int_\Omega u \, \mathrm{d}x$ is the average of $u$.*

To prove one of our results we will also need the following generalization of Poincaré's inequality.

**Theorem B.2** (Maz'ya (2011) Section 6.11.1, corollary 2). *Let $\Omega \subseteq \mathbb{R}^n$ be an open set with finite volume, $u \in W^{1,p}(\Omega)$ such that the trace of $u$ on the boundary $\partial\Omega$ is $r$-integrable, that is $\|u\|_{L^r(\partial\Omega)} < \infty$. Then, for every $r, p, q$ such that $(n-p)r \le p(n-1)$ and $q = \frac{rn}{n-1}$, it holds*

$$\|u\|_{L^q(\Omega)} \le C \left( \|\mathrm{D}u\|_{L^p(\Omega)} + \|u\|_{L^r(\partial\Omega)} \right) \tag{20}$$

*In particular, if $p = r = 2$ we have that $q = \frac{2n}{n-1}$ but since $\Omega$ has finite volume, the inequality holds for each $q \le \frac{2n}{n-1}$ and, in particular, for $q = 2$.*

**Corollary B.3.** *If $p = r = 2$, theorem B.2 holds for $q = \frac{2n}{n-1}$ and, since $\Omega$ has finite volume, it holds for each $q \le \frac{2n}{n-1}$ and, in particular, for $q = 2$, that is*

$$\|u\|_{L^2(\Omega)} \le C \left( \|\mathrm{D}u\|_{L^2(\Omega)} + \|u\|_{L^2(\partial\Omega)} \right) \tag{21}$$

We are now ready to prove theorem 3.2.

*Proof.* Let $\hat{u}_n$ be a sequence such that $L(\hat{u}_n, u) \le \epsilon_n$ with $\epsilon_n \to 0$ and $v_n = \hat{u}_n - u$. From the definition of $L$ have that $\|\mathrm{D}v_n\|_{L^2(\Omega)} \le \epsilon_n$ and similarly for $\|v_n\|_{L^2(\partial\Omega)}$, so that

$$\begin{aligned}
\|v_n\|_{L^2(\Omega)} &\le C \left( \|\mathrm{D}v_n\|_{L^2(\Omega)} + \|v_n\|_{L^2(\partial\Omega)} \right) \le 2C\epsilon_n \to 0 \\
\|v_n\|_{W^{1,2}(\Omega)} &= \|v_n\|_{L^2(\Omega)} + \|\mathrm{D}v_n\|_{L^2(\Omega)} \le 2(C+1)\epsilon_n \to 0
\end{aligned} \tag{22}$$

which gives us the first part of the thesis. Additionally, $\mathrm{D}v_n \to 0$. This means that the limit of $v_n$ is a.e. constant and, actually, continuously differentiable in $\Omega$. Since $v_n \to 0$ and $\lim_n \mathrm{D}v_n = \mathrm{D}v_\infty$ (limits and weak derivatives commute), at the limit $\hat{u}_\infty = u$ and, by continuity of the trace operator in $W^{1,2}(\Omega)$, we also have that $\hat{u}_\infty|_{\partial\Omega} = u|_{\partial\Omega}$ in $L^2(\partial\Omega)$. The result can be easily extended to multi-component functions by considering each component individually. $\square$

### B.3 Proof of theorem 3.3

*Proof.* We show the results for time-independent PDEs; for time-dependent PDEs, it is sufficient to consider the extended domain $\tilde{\Omega} = [0, T] \times \Omega$, in which the initial (and final) conditions become boundary conditions. We also show the result for functions with one component. The result can be generalized by applying it to each component.

Let $\hat{u}$ be the trained Neural network. From theorem 3.2, we have that a network $\hat{u}$ trained with the loss in equation 6 with $m$ derivative terms converges to the solution of the PDE $u$ in $W^{m,2}(\Omega)$. This implies that, when $L(\hat{u}, u) \to 0$, all the derivatives of $\hat{u}$ converge to the true ones $D^l \hat{u} \to D^l u$ for $l = 0, \dots, m$. Since $\mathcal{F}$ is continuous, it follows that $\mathcal{F}[\hat{u}] \to \mathcal{F}[u] = 0$. As a consequence, DERL learns to fulfill the underlying PDE of the system. $\qquad \square$

*Remark* B.4. In the case of time-dependent PDEs, we will not provide the final conditions at $t = T$, which are part of the "boundary" of the extended domain and are necessary for Theorem 3.3. Since our experiments are on regular functions, this has proven not to be an issue, and the neural network still converges.

### B.4 Proof of theorem 3.6

*Proof.* Assuming the result for smooth integrable functions, we show the thesis using the characterization of the Sobolev space $W^{1,2}(\mathbb{R}^n)$ via $C_c^\infty(\mathbb{R}^n)$ approximations with smooth functions with compact support (Maz'ya, 2011). For each $\delta > 0$, there exists $\phi \in C_c^\infty(\mathbb{R}^n)$ such that $\|u - \phi\|_{W^{1,2}(\mathbb{R}^n)} < \delta$. First, we note that for each $u \in W^{1,2}(\mathbb{R}^n)$

$$
\begin{aligned}
|u(\boldsymbol{x} + h\boldsymbol{e}_i) - u(x)| &= \left| \int_0^1 \frac{\partial u}{\partial x_i}(\boldsymbol{x} + ht\boldsymbol{e}_i) h \, \mathrm{d}t \right| \\
&\leq \int_0^1 \left| \frac{\partial u}{\partial x_i}(\boldsymbol{x} + ht\boldsymbol{e}_i) \right| |h| \, \mathrm{d}t
\end{aligned}
\tag{23}
$$

so that, by squaring and integrating

$$
\left\| \frac{u(\boldsymbol{x} + h\boldsymbol{e}_i) - u(\boldsymbol{x})}{h} \right\|_{L^2(\mathbb{R}^n)}^2 \leq \left\| \int_0^1 \left| \frac{\partial u}{\partial x_i}(\boldsymbol{x} + ht\boldsymbol{e}_i) \right| \mathrm{d}t \right\|_{L^2(\mathbb{R}^n)}^2 \leq \left\| \frac{\partial u}{\partial x_i} \right\|_{L^2(\mathbb{R}^n)}^2,
\tag{24}
$$

which is the first part of the thesis. Applying this to $u - \phi$ one directly shows that

$$
\|\mathrm{D}_{x_i}^\epsilon u - \mathrm{D}_{x_i}^\epsilon \phi\|_{L^2(\mathbb{R}^n)}^2 \leq \|u_{x_i} - \phi_{x_i}\|_{L^2(\mathbb{R}^n)},
\tag{25}
$$

Then, we have that

$$
\begin{aligned}
\|\mathrm{D}_{x_i}^\epsilon u - u_{x_i}\|_{L^2(\mathbb{R}^n)} &= \|\mathrm{D}_{x_i}^\epsilon u - \mathrm{D}_{x_i}^\epsilon \phi + \mathrm{D}_{x_i}^\epsilon \phi - \phi_{x_i} + \phi_{x_i} - u_{x_i}\|_{L^2(\mathbb{R}^n)} \\
&\leq \|\mathrm{D}_{x_i}^\epsilon u - \mathrm{D}_{x_i}^\epsilon \phi\|_{L^2(\mathbb{R}^n)} + \|\mathrm{D}_{x_i}^\epsilon \phi - \phi_{x_i}\|_{L^2(\mathbb{R}^n)} + \|\phi_{x_i} - u_{x_i}\|_{L^2(\mathbb{R}^n)} \\
&\leq \|u_{x_i} - \phi_{x_i}\|_{L^2(\mathbb{R}^n)} + \delta + \delta \\
&\leq 3\delta
\end{aligned}
\tag{26}
$$

where the second inequality follows from equation 25 and the last one follows from choosing a small enough $\epsilon$. The same steps are true for every open subset $\Omega$, by considering the $L^2_{\mathrm{loc}}$ norm. $\qquad \square$

## C Additional Results and Experiments for Section 4

In this Section, we provide additional results and experiments. For the complete list of tasks see table 8.

### C.1 Allen-Cahn equation

We now present the additional results for the Allen-Cahn equation. Figure 6a shows a direct comparison of the errors on the learned solution $u$ in the domain while figure 6b shows the error difference for the PDE

Table 8: Summary of the tasks.

| Experiment | Equation | Description | Tasks |
|---|---|---|---|
| **Allen-Cahn** | (AC)   $\lambda(u_{xx} + u_{yy}) + u(u^2 - 1) = f$ | Second-order PDE | Generalize in-domain
Generalize to new parameters
Transfer to a larger domain
Transfer to new parameters |
| **Continuity** | (CO)   $\frac{\partial u}{\partial t} + \nabla \cdot (\boldsymbol{v}u) = 0$ | Time-dependent PDE | Generalize in-domain |
| **Pendulum** | (DP)   $\ddot{u} + \frac{g}{l}\sin(u) + \frac{b}{m}\dot{u} = 0$ | ODE | Generalize to new parameters |
| **KdV distillation** | (KdV)   $u + uu_t + \nu u_{xxx} = 0$ | Third-order PDE | Higher-order distillation |
| **Navier-Stokes** | (NS.M)   $[D\boldsymbol{u}] \cdot \boldsymbol{u} - \Delta\boldsymbol{u} + \nabla p = 0$
(NS.I)$\nabla \cdot \boldsymbol{u} = 0$ | System of PDEs (Appendix) | Generalize in-domain |
| **NCL distillation** | (A2.C)   $\frac{\partial \rho}{\partial t} + \nabla \cdot (\rho\boldsymbol{u}) = 0$
(A2.I)$\nabla \cdot \boldsymbol{u} = 0$
(A2.M)   $\frac{\partial \boldsymbol{u}}{\partial t} + [D\mathbf{u}]\boldsymbol{u} + \frac{\nabla p}{\rho} = 0$ | System of PDEs (Appendix) | Distillation on
different architecture |

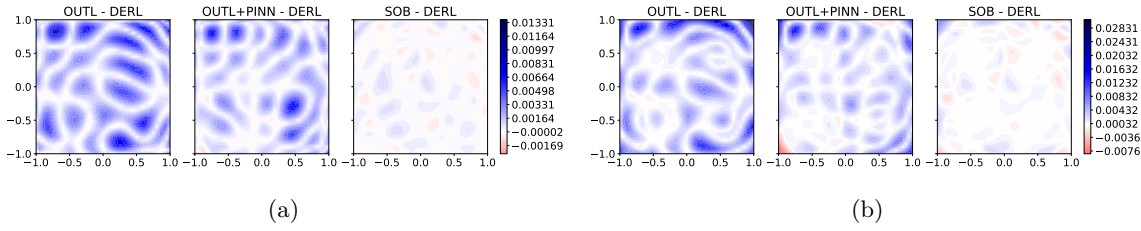

(a)                                           (b)

Figure 6: Allen-Cahn experiment: (a) $u$ error comparison between DERL and the other methodologies. (b) PDE residual comparison between DERL and the other methodologies. Blue regions are where DERL performs better.

residual. We computed the local error on grid points for each method. Then, for each method, we computed the difference between its errors and DERL's error, such that positive values (blue) are where DERL performs better. These pictures show clearly the performance of DERL and the improvements in learning a physical system with its partial derivatives.

**Experiment on an equispaced grid.** We also experimented with learning from an equispaced grid of collocation points with $\mathrm{d}x = 0.02$ and the ADAM optimizer (Kingma, 2014). Table 9 reports the metrics for this setup. For a graphical comparison, in figure 7 we report the $L^2$ error on $u$ and PDE residual in the form of differences between the other methodologies and DERL: blue regions are where we perform better. The key takeaway is, as in Section 4.1, that learning the derivatives is both sufficient and better than learning just $u$, even with this setup. DERL and SOB performed best in all metrics, while OUTL and its combination with PINN are at least 2 times worse.

Table 9: Results for the Allen-Cahn equation on grid points with Adam optimizer.

| Model | $\|u - \hat{u}\|_2$ | $\|\mathcal{F}[\hat{u}]\|_2$ |
|---|---|---|
| **DERL** (ours) | **0.01235** | **0.01173** |
| **OUTL** | 0.03258 | 0.02157 |
| **OUTL+PINN** | 0.02109 | 0.02896 |
| **SOB** | 0.01500 | 0.01578 |

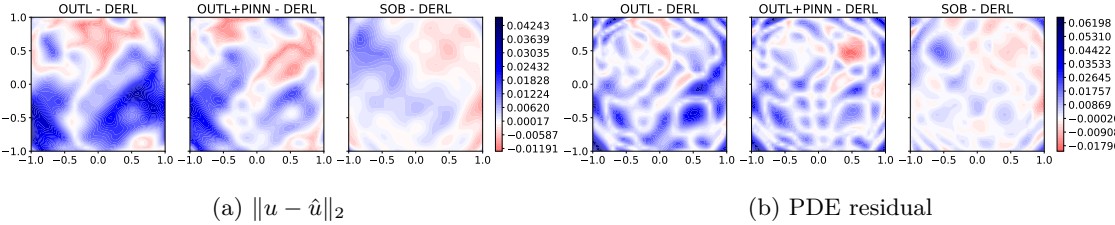

(a) $\|u - \hat{u}\|_2$                    (b) PDE residual

Figure 7: Graphical results for the Allen-Cahn experiment with grid points and Adam optimizer. $L^2$ loss on $u$ and PDE residual comparison between the methodologies. Differences between the methodology and the DERL errors. Positive (blue) regions are where we perform better.

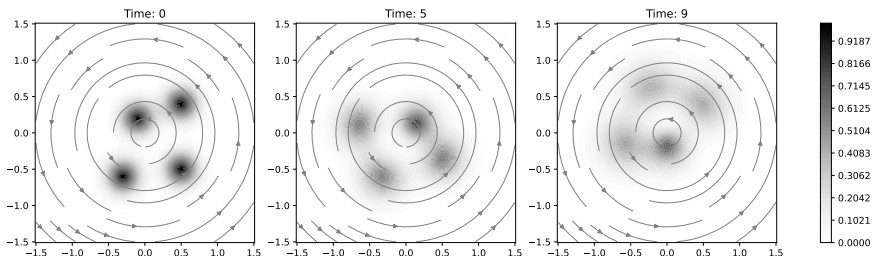

Figure 8: Solution to the continuity equation at $t = 0, 5, 9$

## C.2  Continuity equation

**Setup.** The original solution was calculated using finite volumes (Ferziger & Peric, 2001) on a grid with $\Delta x = \Delta y = 0.01$ in the domain $[-1.5, 1.5]$ and time discretization of $\Delta t = 0.001$ for $t \in [0, 10]$. For the points used during training, we downsampled the full grid by a factor of 10 for each spatiotemporal coordinate. In the testing phase, we test the model on the full grid, without downsampling. This way, we can test which model can best learn the solution on a finer grid, while being consistent with the equation. The empirical derivatives are calculated with $h = 0.01$ on each component.

We start by showing the true solution to the continuity equation at $t = 0, 5, 9$ in figure 8, together with the $L^2$ errors at $t = 0$ (figure 9a), $t = 5$ (figure 9b), and $t = 9$ (figure 9c). The results show clearly that the best methodologies to learn the density are DERL and OUTL, with the latter being the worst at physical consistency as reported in Section 4.1. For this task, we performed an additional experiment with a PINN. We found similar results to Wang et al. (2022a), where the model deviates from the true solution as time increases. We notice that DERL, while not having access to the density $\rho$ at later times, does not show such behavior.

**Randomly sampled points.** For this equation, we performed an additional experiment where we tested the effects of using an interpolated curve on the solution to randomly sample points and calculate derivatives. This simulates a situation with sparse data and, as above, no analytic derivatives of $u$ available. For this purpose, we used the data from the grid calculated as in Section 4.1 and used a regular grid interpolator from SciPy (Virtanen et al., 2020) with cubic interpolation for third-order accuracy. Then, we randomly sampled 10000 points in the $t, x, y$ domain, and calculated finite difference derivatives with $h = 0.001$.

Numerical results are available in table 10, while figures 10b and 10a show the comparisons on physical consistency (PDE $L^2$ residual) and the $L^2$ error on $\rho$ across methodologies. The format is the usual difference between the methods' error and the DERL one, with positive values (blue regions) meaning we are performing better. Even in this case DERL is among the best-performing models, with a $u$ error and physical consistency respectively similar to SOB. OUTL is clearly the worst at learning the physics of the problem, even in combination with the PINN loss. The PINN alone shows again a discrepancy from the true solution that increases with time. This shows that our methodology works even with empirical derivatives calculated on an

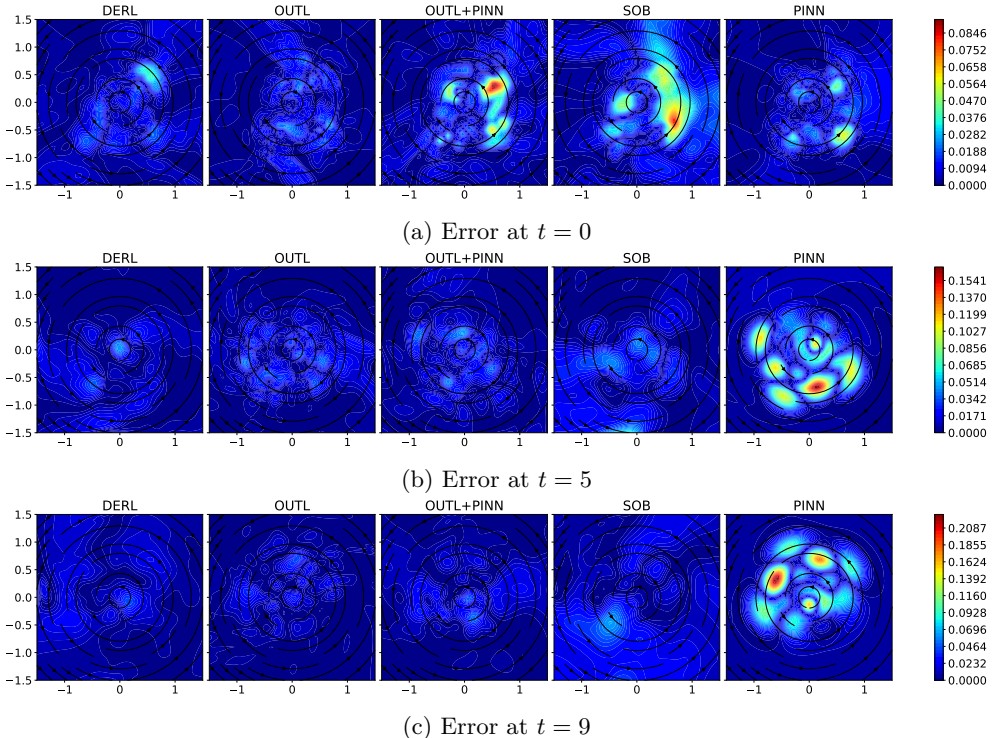

(a) Error at $t = 0$

(b) Error at $t = 5$

(c) Error at $t = 9$

Figure 9: Continuity equation experiment. (a) True densities at $t = 0, 5, 9$. $L^2$ errors for the compared methodologies at (b) $t = 0$, (c) $t = 5$, (d) $t = 9$.

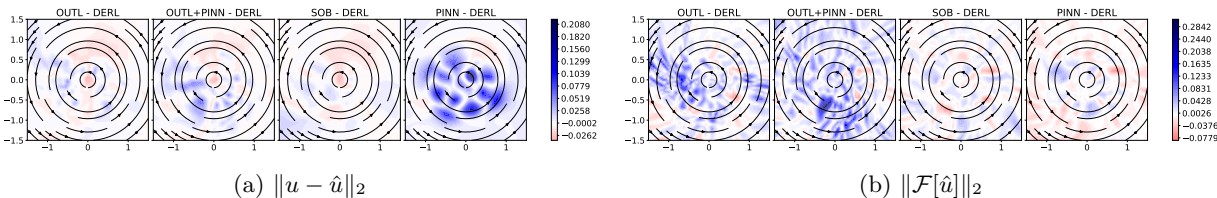

(a) $\|u - \hat{u}\|_2$      (b) $\|\mathcal{F}[\hat{u}]\|_2$

Figure 10: Continuity equation task with randomly sampled points. Differences between the other methods' residuals and DERL at $t = 5$. Blue regions are where we perform better.

Table 10: Numerical results for the continuity equation with randomly sampled points.

| Model | $\|u - \hat{u}\|_2$ | $\|\mathcal{F}[\hat{u}]\|_2$ |
|---|---|---|
| **DERL** (ours) | 0.08714 | 0.1948 |
| **OUTL** | 0.07308 | 0.4065 |
| **OUTL+PINN** | 0.1390 | 0.4393 |
| **SOB** | **0.07190** | **0.1815** |
| **PINN** | 0.3296 | 0.1499 |

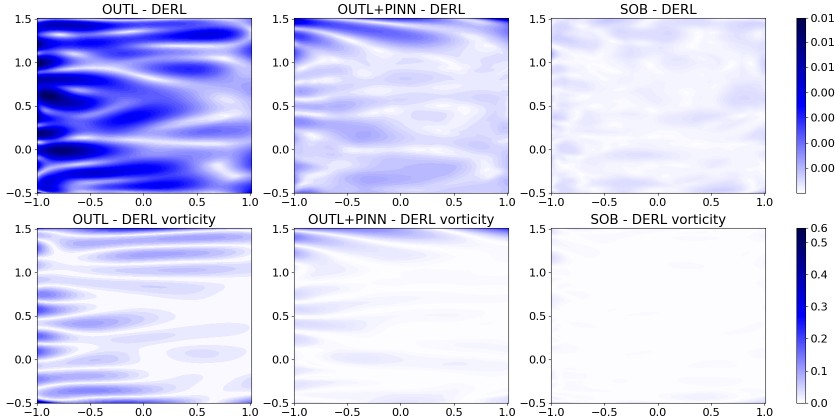

Figure 11: Error difference plot for the Kovasznay flow experiment on the Navier-Stokes equation. The first row is the local error on the solution $(\boldsymbol{u}, p)$, while the second row is the error on the vorticity $\omega$. Each point is calculated as the difference between the error by the model and the error by DERL. Blue regions are where DERL performs better.

interpolation of the true solution. We remark again on the importance of learning the derivatives, as DERL and SOB are the best all-around models for this task.

### C.3    Navier-Stokes: Kovasznay flow

The analytical solution to the Kovasznay flow task is

$$\boldsymbol{u}(x,y) = \left[1 - e^{\lambda x}\cos(2\pi y), \frac{\lambda}{2\pi}e^{\lambda x}\sin(2\pi y)\right]^{\top}, \qquad p(x,y) = \frac{1}{2}\left(1 - e^{2\lambda x}\right), \tag{27}$$

where $\lambda = \frac{1}{2\nu} - \sqrt{\frac{1}{4\nu^2} + 4\pi^2}$. In this case, the vorticity is given by $\omega = \frac{\lambda}{\nu}e^{\lambda x}\frac{\sin(2\pi y)}{2\pi}$. Figure 11 shows the error difference between DERL and the other methodologies, both for the solution $(\boldsymbol{u}, p)$ and the vorticity $\omega$.

### C.4    Pendulum

We start by describing the pendulum problem from a physical point of view. We consider the rope pendulum under the force of gravity $F_g = -mg$ and a dampening force $F_d = -bl\dot{u}$, where $u(t)$ is the angle the rope makes w.r.t. the vertical, $b$ is a parameter for the dampening force, $l, m$ are respectively the length of the rope and the mass of the pendulum. A schematic representation of the system can be found in figure 12a, and we also plot its phase space on $(u, \dot{u})$ along with some trajectories with dampening 12b or without 12c, that is the conservative case. The evolution of the pendulum is given by the ODE $\mathcal{F}[u] = ml\ddot{u} + mg\sin(u) + bl\dot{u} = 0$

#### C.4.1    Differentiability with respect to Initial Conditions

For these kinds of ODEs, the solution $(u(t, u_0, \dot{u}_0), \dot{u}(t, u_0, \dot{u}_0))$ is actually continuously differentiable with respect to the IC $(u_0, \dot{u}_0)$. For a proof, see Hartman (2002), theorem 3.1. In this specific case, we can derive the following PDEs:

$$\mathcal{G}[u] := \begin{cases} \dfrac{\partial}{\partial u_0}\dfrac{\partial^2 u}{\partial t^2} + \dfrac{g}{l}\dfrac{\partial \sin(u)}{\partial u_0} + \dfrac{b}{m}\dfrac{\partial}{\partial u_0}\dfrac{\partial u}{\partial t} = 0, \\[3mm] \dfrac{\partial}{\partial \dot{u}_0}\dfrac{\partial^2 u}{\partial t^2} + \dfrac{g}{l}\dfrac{\partial \sin(u)}{\partial \dot{u}_0} - \dfrac{b}{m}\dfrac{\partial}{\partial \dot{u}_0}\dfrac{\partial u}{\partial t} = 0 \end{cases} \tag{28}$$

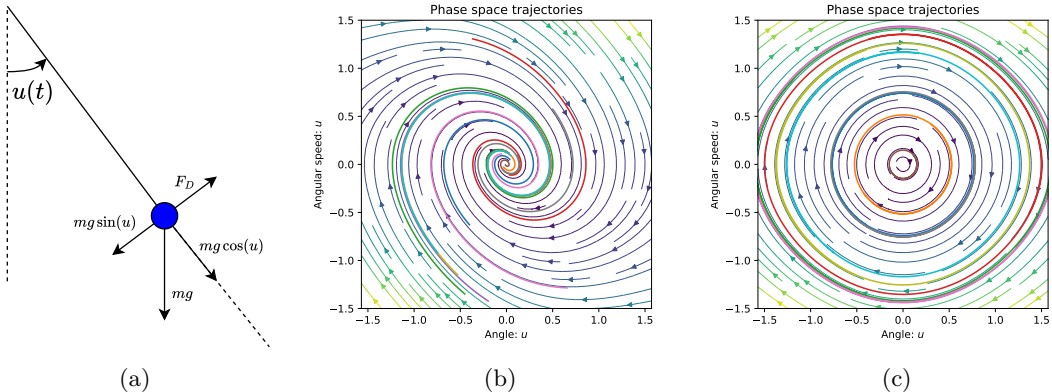

Figure 12: (a) Schematic representation of the pendulum system: the state variables are the angle $u$ the rope makes with the vertical direction. (b) Phase space of the damped pendulum. Arrows represent the direction and intensity of $\dot{u}, \dot{\omega}$. Testing trajectories are also plotted. (c) Phase space of the conservative pendulum with no dampening ($b = 0$).

which are true for any trajectory $(u(t, u_0, \dot{u}_0), \dot{u}(t, u_0, \dot{u}_0))$. In the experiments, we calculated the $L^2$ PDE residual of equation 28 in plain PINN style, to see which model learns to be differentiable w.r.t. the IC.

### C.4.2 HNN and LNN setup

In this Section, we describe the setup we used for the Hamiltonian Neural Network (Greydanus et al., 2019) and Lagrangian Neural Network (Cranmer et al., 2019). In particular, we will explain how we adapted these methods to learn non-conservative fields, which is not present in their original works.

In both the conservative and dampened cases, we used the setup described in their original articles without modifications in the architecture, initialization, and training. We now describe how to include the dampening effects through data or particular processing during inference.

**Hamiltonian Neural Network.** In this case, the neural network parametrizes the Hamiltonian function $H(p, q)$, from which derivatives one obtains

$$\frac{\partial p}{\partial t} = -\frac{\partial H}{\partial q}, \qquad \frac{\partial q}{\partial t} = \frac{\partial H}{\partial p}. \tag{29}$$

Equation 29 is valid in the conservative case, while in the dampened case it is sufficient to add the term $-bp$ to $\frac{\partial p}{\partial t}$, leading to the correct formulation as in equation (DP) in table 8. This way, the HNN learns just the conservative part of the field, which was originally made for, while we add the dampening contribution externally.

**Lagrangian Neural Network.** In this case, we need to look at the Lagrangian formulation of mechanics, in particular at the Euler-Lagrangian equation. Given a neural network that parametrizes the Lagrangian function $\mathcal{L}(u, \dot{u})$, we have that (see Cranmer et al. (2019) for the full explanation)

$$\frac{\mathrm{d}}{\mathrm{d}t} D_{\dot{u}} \mathcal{L} - D_u \mathcal{L} = 0. \tag{30}$$

We can add the effect of external forces, in this case, the dampening, using d'Alembert's principle of virtual work (Goldstein, 2011). In particular, given the force in vector form

$$\mathbf{F}_d = -bl\dot{u}[\cos(u), \sin(u)] \tag{31}$$

and the position of the pendulum as a function of the coordinates

$$\boldsymbol{x}(t) = [l\sin(u), -l\cos(u)], \tag{32}$$

Table 11: Results for the damped pendulum experiment with randomly sampled times and empirical derivatives.

| Model | $\|u - \hat{u}\|_2$ | $\|\dot{u} - \hat{\dot{u}}\|_2$ | $\|\mathcal{F}[\hat{u}]\|_2$ | $\|\mathcal{G}[\hat{u}]\|_2$ | $\|\mathcal{F}[\hat{u}]_{t=0}^{\text{full}}\|_2$ |
|---|---|---|---|---|---|
| **DERL** (ours) | **0.01237** | **0.01504** | **0.01581** | **0.1106** | **0.4678** |
| **OUTL** | 0.01469 | 0.07637 | 0.07148 | 0.2466 | 1.001 |
| **OUTL+PINN** | 0.01495 | 0.06993 | 0.06588 | 0.2877 | 0.8437 |
| **SOB** | 0.01532 | 0.02983 | 0.02890 | 0.1276 | 0.8365 |

Table 12: Results for the conservative pendulum experiment.

| Model | $\|u - \hat{u}\|_2$ | $\|\dot{u} - \hat{\dot{u}}\|_2$ | $\|\mathcal{F}[\hat{u}]\|_2$ | $\|\mathcal{G}[\hat{u}]\|_2$ | $\|\mathcal{F}[\hat{u}]_{t=0}^{\text{full}}\|_2$ |
|---|---|---|---|---|---|
| **DERL** (ours) | 0.07079 | **0.08459** | **0.04233** | **0.35976** | **0.5577** |
| **OUTL** | 0.1066 | 0.2856 | 0.2510 | 1.865 | 0.8998 |
| **OUTL+PINN** | 0.07638 | 0.1643 | 0.1296 | 0.7740 | 0.8422 |
| **SOB** | **0.06707** | 0.1031 | 0.06986 | 0.5090 | 0.7970 |

the virtual work is given by

$$Q(t, u, \dot{u}) = \mathbf{F}_d \cdot \frac{\partial \boldsymbol{x}(t)}{\partial u} = -bl^2 \dot{u}. \tag{33}$$

It is then sufficient to put this term on the right-hand side of equation 30 and proceed with the calculations as in Cranmer et al. (2019) to obtain the full update of the model.

### C.4.3 Results with empirical derivatives

We repeated the same experiment of Section 4.1, this time using empirical derivatives calculated by using finite difference with $h = 10^{-3}$. We also used randomly sampled times to evaluate trajectories, to simulate a setting where the time sampling is not constant. We report the relevant losses in table 11, where DERL outperforms the other methods in every metric, especially in those showing the generalization capability. We conclude that learning derivatives improves generalization in the models while achieving SOTA results in predicting the trajectories.

### C.4.4 Conservative pendulum

The setup for the conservative pendulum experiment is the same as for the dampened one, with the fundamental difference that $b = 0$ in both data generation and during training. We report the numerical results in table 12, with the same definitions as in Section 4.2. We also plot the field error differences in figure 13. DERL is the best model in the conservative case as well, since it achieves the lowest error in almost all metrics. The only exception is the error on testing trajectories, where it is still second best and very close to SOB. In general, no other method shows the generalization capabilities of DERL, especially in terms of PDE residuals and regularity with respect to initial conditions.

In this case, HNN (field $L^2$ error 0.13570) and LNN (field $L^2$ error 0.085388) outperform all other methodologies as they are precisely built for this task. We still remark that their objective is solely to learn the field, which is easier than whole trajectories.

## C.5 Parametric Allen-Cahn Equation

We provide additional information on the Allen-Cahn parametric experiments, together with additional results.

### C.5.1 Experiment with One parameter

**Setup.** The MLPs have 4 hidden layers with 50 units each, similar to Raissi et al. (2019). The models take as input the coordinate and parameter of the equation $(x, y, \xi)$ and output the solution for the Allen-Cahn

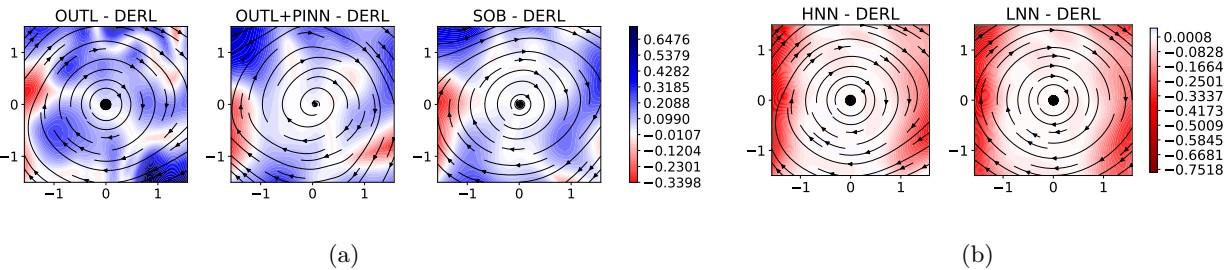

Figure 13: Comparison in the learned field at $t = 0$ for the pendulum experiment expressed as the differences of the $L^2$ errors between the other methodologies and DERL. The blue area is where we perform is better.

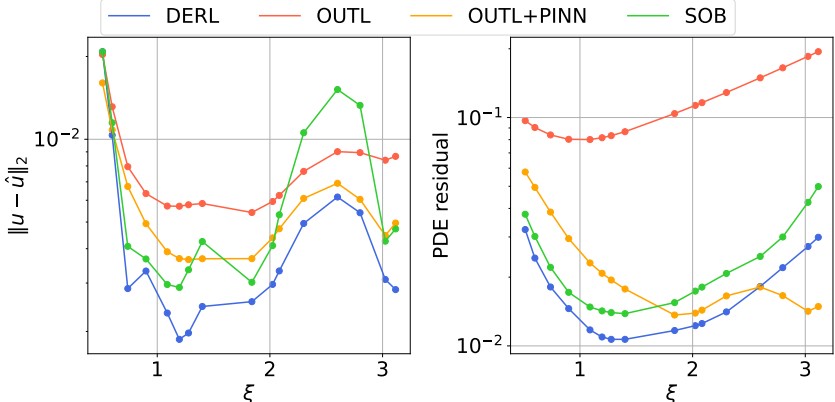

Figure 14: Allen Cahn parametric experiments: Single parameter equation. $u$ error (left) and PDE residual (right) as functions of $\xi$.

equation with parameter $\xi$, that is $\hat{u}(x, y; \xi)$. The networks are trained with batch size 100 with the Adam optimizer for 50 epochs, then are further optimized for another 50 epochs with the BFGS optimizer on the whole dataset, similarly to Penwarden et al. (2023). The solution on 1000 random collocation points is calculated for 16 values of $\xi$. A train/validation/test split is applied, with 8 parameters for training, 2 for validation, and 6 for testing. Testing values of $\xi$ are chosen to explore regions of the parameter space not seen during training and validation. The training and validation datasets are used for tuning the hyperparameters of the models, that is the learning rate and the weights of the loss terms in equation 17.

**Results.** Numerical results are provided in Section 4.2. Figure 14 shows the error on $u$ and the PDE residual of each model for different values of the parameters. As we can see, DERL is consistently the best model. We also provide the comparison plots for the solution $u$ on the testing $\xi$ values in figure 15. As before, the plots show a comparison between DERL and the other methods' errors, such that blue regions are where DERL performs better. It is clear also from a visual point of view that DERL is the best model to predict $u$ and to simultaneously have a low PDE residual, especially compared to OUTL.

### C.5.2 Experiment with Two Parameters

This Section provides additional results and information on the setup for the double parametric Allen-Cahn experiment in Section 4.2. The MLPs have 4 hidden layers with 50 units each, similar to Raissi et al. (2019). The models take as input the coordinate and parameters of the equation $(x, y, \xi_1, \xi_2)$ and output the estimate of $u$ with parameters $(\xi_1, \xi_2)$, that is $\hat{u}(x, y; \xi_1, \xi_2)$. The networks are trained with batch size 100 with the Adam optimizer for 100 epochs, then are further optimized for another 100 epochs with the BFGS optimizer on the whole dataset, similarly to Penwarden et al. (2023). The solution on 1000 random collocation points

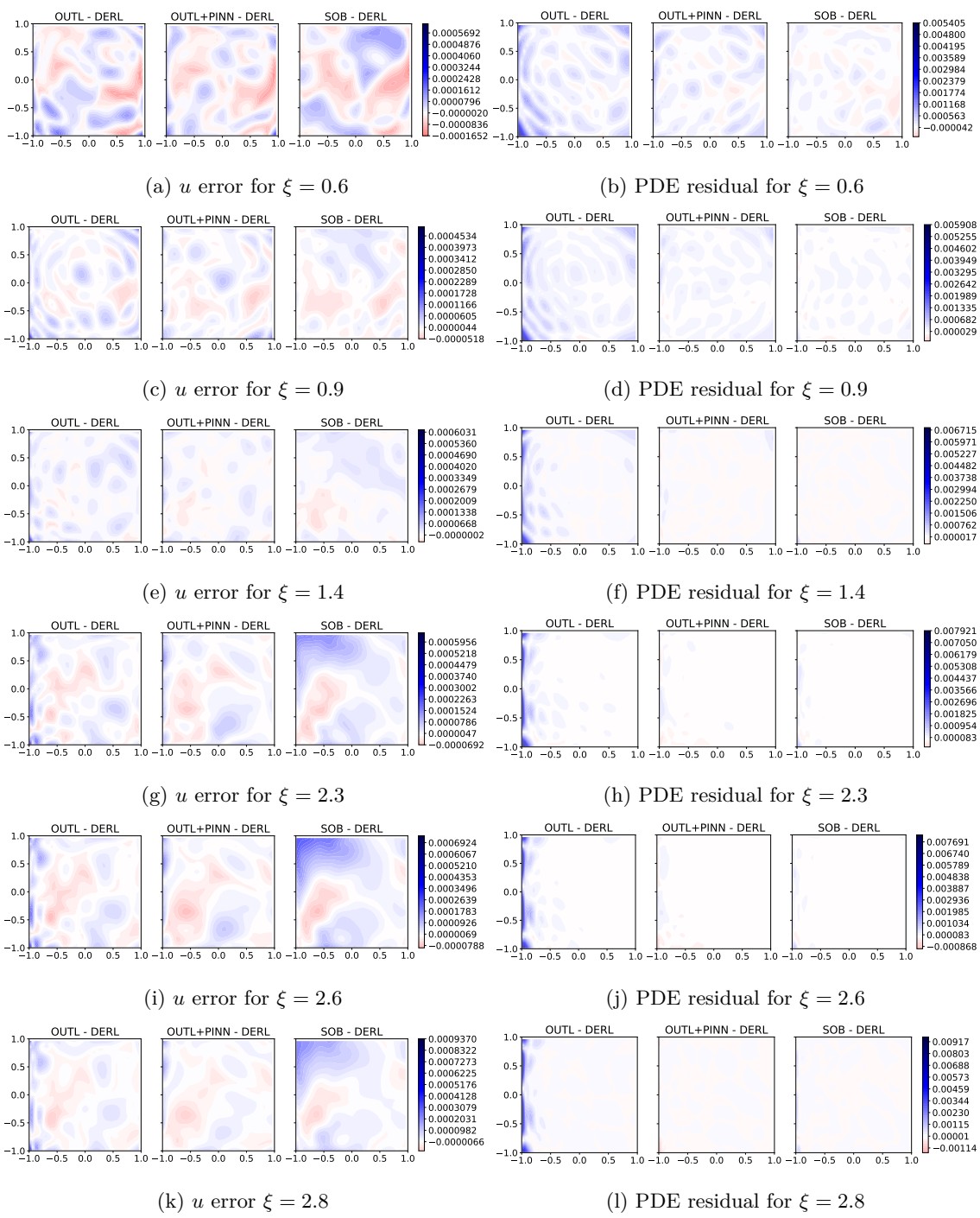

(a) $u$ error for $\xi = 0.6$           (b) PDE residual for $\xi = 0.6$

(c) $u$ error for $\xi = 0.9$           (d) PDE residual for $\xi = 0.9$

(e) $u$ error for $\xi = 1.4$           (f) PDE residual for $\xi = 1.4$

(g) $u$ error for $\xi = 2.3$           (h) PDE residual for $\xi = 2.3$

(i) $u$ error for $\xi = 2.6$           (j) PDE residual for $\xi = 2.6$

(k) $u$ error $\xi = 2.8$           (l) PDE residual for $\xi = 2.8$

Figure 15: Comparison plots for the solution and the forcing term in the Allen-Cahn parametric equation with one parameter on test values of $\xi$

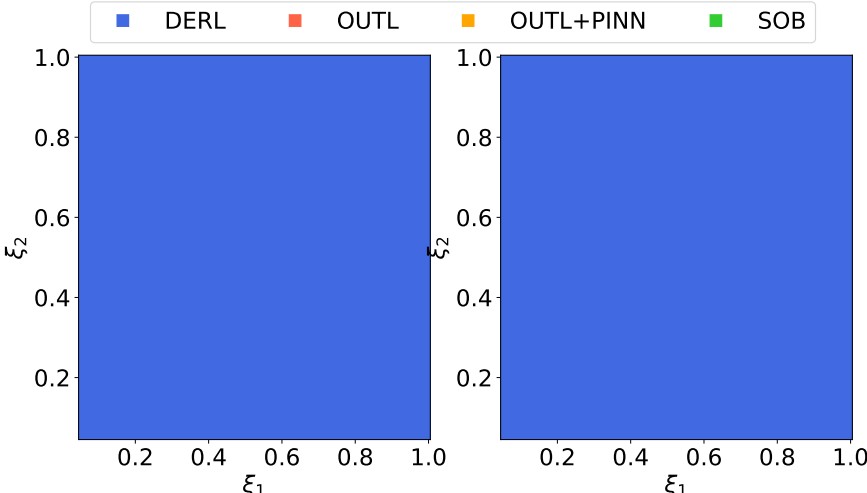

Figure 16: Double-parameter Allen-Cahn equation. The plot is obtained by taking a grid of points $\xi_1, \xi_2$ in the parameter space and coloring it with the color of the model with the lowest $u$ error or PDE residual for such parameters.

is calculated for 25 couples of $(\xi_1, \xi_2)$. A train/validation/test split is applied, with 15 couples of parameters for training, 5 for validation, and 5 for testing. The training and validation datasets are used for tuning the hyperparameters of the models, that is the learning rate and the weights of the loss terms in equation 17.

Figure 16 shows the parameters' space colored with a color code representing the best model in that region. In particular, each point in the plot represents a couple of parameters, which are colored with the color of the model achieving the lowest $u$ error and PDE residual for that couple of parameters. DERL is the best in the whole domain.

Similarly to before, figure 17 shows the error comparisons on $u$ and the PDE residuals for the 5 testing couples $(\xi_1, \xi_2)$. Figures clearly show that DERL outperforms the other models.

# D    Additional Results and Experiments for Section 5

## D.1    Higher Order Derivatives

**Teacher and student model setup.**    We consider the Korteweg-de Vries (KdV) equation, a third-order non-linear PDE (equation (KdV) in table 8), with $\nu = 0.0025$. The IC is $u(0, x) = \cos(\pi x)$ with periodic BC for $u$ and $u_x$. The reference solution is first obtained using the Scipy (Virtanen et al., 2020) solver with Fast Fourier Transforms on a grid with $\Delta x = \Delta t = 0.005$. Then, a PINN is carefully trained on equation (KdV) in table 8 and is treated as the teacher model to distill knowledge from. For the teacher model, we used an MLP with 9 layers of 50 units, tanh activation, trained for 200 epochs with batch size 64 and Adam optimizer. The higher number of layers is applied due to the complexity of the KdV task, as in Jagtap et al. (2020). After the teacher's training, we save a dataset of its outputs $\hat{u}_T(x, t)$, its derivatives $D\hat{u}_T(x, t)$, and its hessian matrix $D^2\hat{u}_T(x, t)$ on the training collocation points. As always, partial derivatives are calculated via AD Baydin et al. (2018). Then, the student networks are trained with the specific distillation loss of the method, e.g. $D\hat{u}_S - D\hat{u}_T$ for DERL, and the original BC and IC. Since this is a third-order PDE, we are also interested in understanding the impact of higher-order derivatives on the performance, both in terms of the PDE residual and in terms of matching the true solution $u$ or the BC. We implemented **Hessian learning (HESL)**, which learns the Hessian matrices $D^2\hat{u}_T$ of the teacher model. We also explored its combination with DERL (DER+HESL, which learns both $D\hat{u}_T$ and $D^2\hat{u}_T(x, t)$) and Sobolev (SOB+HESL). In this last case, we use the approximation of the Hessian matrix as suggested in Czarnecki et al. (2017), while for HESL we use the full Hessian of the network, at the cost of an increase in computational time (see Section A.1 for

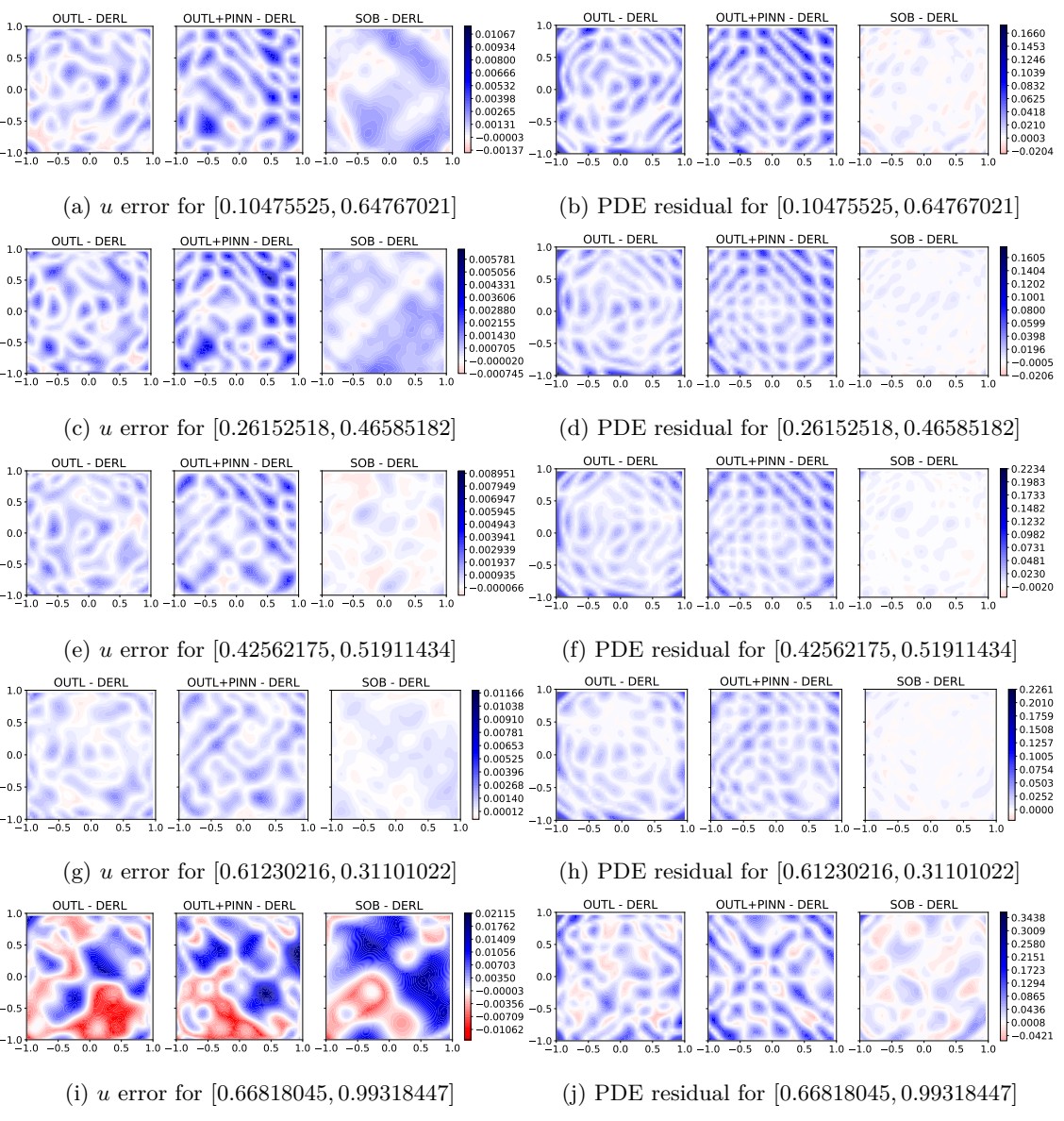

Figure 17: Comparison plots for the solution and the forcing term in the Allen-Cahn parametric equation with two parameters on test couples $(\xi_1, \xi_2)$

Table 13: Results for the KdV equation, PINN distillation. Results are empirical $L^2$ norms over the time-space domain. Derivative and Hessian losses are computed with respect to the PINN. Best model(s) in bold, second best underlined.

| Model | $\|\hat{u}_S - u\|_2$ | $\|\mathrm{D}\hat{u}_S - \mathrm{D}\hat{u}_T\|_2$ | $\|\mathrm{D}^2\hat{u}_T - \mathrm{D}^2\hat{u}_S\|_2$ | $\|\mathcal{F}[\hat{u}_S]\|_2$ | BC loss |
|---|---|---|---|---|---|
| PINN (teacher) | 0.037171 | / | / | 0.16638 | 0.33532 |
| **DERL** (ours) | 0.038331 | 0.098188 | 3.9872 | 0.32480 | 0.014197 |
| **HESL** (ours) | **0.037380** | 0.065454 | 1.1662 | **0.19153** | 0.014220 |
| **DER+HESL** (ours) | 0.038524 | **0.041988** | **0.85280** | 0.19317 | 0.031850 |
| **OUTL** | 0.038589 | 0.22580 | 31.582 | 17.366 | **0.012830** |
| **SOB** | 0.037447 | 0.10097 | 4.0967 | 0.38523 | 0.013644 |
| **SOB+HES** | 0.041353 | 0.13119 | 3.2684 | 0.23184 | 0.016222 |

Table 14: Results for the NCL distillation experiment. $L^2$ distance between the NCL and our distilled solutions, $L^2$ norms of the residuals of the 2 PDEs. Norms are calculated across the entire time-space domain. The best results are in bold.

| Model | $\|\hat{u}_T - \hat{u}_S\|$ | (A2.M) $\|\mathcal{F}[\hat{u}_S]\|$ | (A2.I) $\|\mathcal{F}[\hat{u}_S]\|$ |
|---|---|---|---|
| **DERL** (ours) | 0.015287 | **0.28566** | **0.095044** |
| **OUTL** | 0.022247 | 0.52101 | 0.17159 |
| **SOB** | **0.013282** | 0.28620 | 0.10134 |

further details). Each methodology has been tuned individually to find the hyperparameters that provide the best approximation of the true teacher's solution $\hat{u}_T$.

**Results.** Table 13 shows the relevant metrics. Together with the usual ones, we also report the $L^2$ distances between the teacher's partial derivatives and the student's ones, that is $\|\mathrm{D}\hat{u}_T - \mathrm{D}\hat{u}_S\|_2$ and $\|\mathrm{D}^2\hat{u}_T - \mathrm{D}^2\hat{u}_S\|_2$, and the BC loss. First, we notice that all methods successfully learned to approximate the solution $u$ with the same performance as the teacher model, with SOB+HESL being the worst by a small margin. We observe that (a) distilling a PINN can lead to noticeable **performance improvements**. In particular, our HESL and DER+HESL achieved the same PDE residual and distance to the true solution of the teacher model but with a BC loss smaller by at least one order of magnitude. This may be due to the easier optimization objective of the student models. (b) OUTL fails to distill the physical knowledge of an architecturally identical PINN, as the PDE residual is two orders of magnitude larger than any other model. Interestingly, the best methods in terms of physical consistencies are those that did not see the values of $u$ directly. (c) When boundary conditions are available for both $u, u_x$, **second order derivatives are sufficient** to learn the true solution with PDE consistency comparable to PINN. (d) HESL and DER+HESL showed the best overall results. This tells us that adding one more derivative helps learning and distilling high-order PDEs. Furthermore, approximating Hessians as in SOB+HESL (Czarnecki et al., 2017) slightly reduces the performance compared to using the full Hessian of the network.

Figures 18a and 18b show on the $(t, x)$ domain the comparison of the errors on $u(t, x)$ and the local PDE residual for the tested distillation methods.

As we can see, every tested method learns similarly to approximate the true solution $u(t, x)$, with slightly better performances on models with fewer derivatives to learn. On the other hand, more derivatives help the consistency of the model to the underlying PDE compared to no derivatives, as in OUTL, which fails in this regard. It is worth noticing that the PINN was not able to optimize the BC, while other methodologies adapted well to them.

## D.2  Neural Conservation Laws

We perform knowledge distillation with the NCL architecture (Richter-Powell et al., 2022), considering the Euler equations for incompressible inviscid fluids with variable density in the 3D unit ball (3 PDEs). The system is made of the mass conservation (A2.C), incompressibility (A2.I), and momentum (A2.M) equations

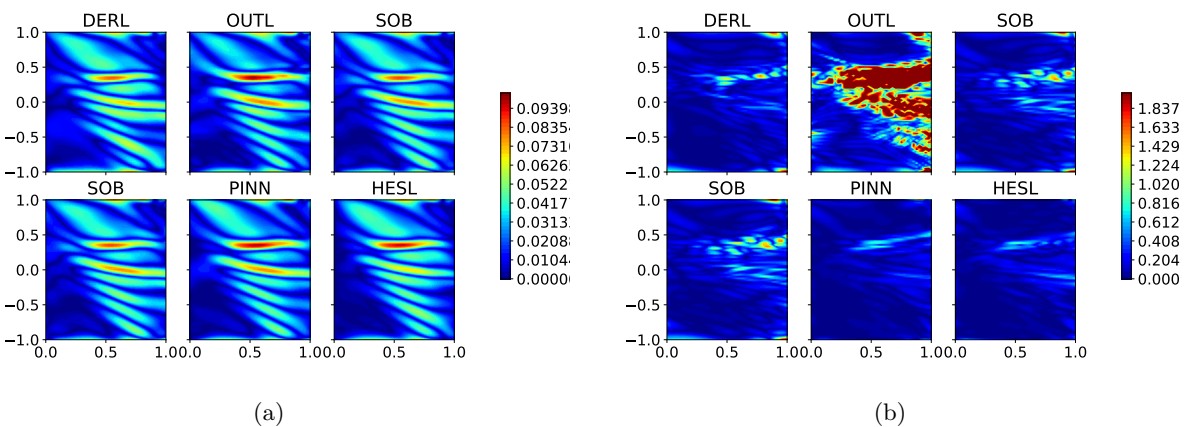

Figure 18: Korteweg-de Vries equation experiment. (a) Errors on the true solution $u$ in the $(t, x)$ space. (b) Local PDE $L^2$ residual. In OUTL, the residual is capped at 2 for clarity reasons on the other plots.

(table 8), where $\rho$ is the density, $\boldsymbol{u}$ the velocity and $p$ the pressure. IC and BC are from Richter-Powell et al. (2022). Equation (A2.D) is guaranteed by the specific architecture of the NCL model, while (A2.M) and (A2.I) are learned with the usual PINN style PDE residual loss. The teacher and student models' setup is the same as in Richter-Powell et al. (2022).

**Model setup.** The model architecture is the same as in Richter-Powell et al. (2022), that is a MLP with 4 layers of 128 units, trained for 10000 steps on batches of 1000 random points in the 3D unit ball. As in the previous task, each student model has the same architecture as the teacher model, and the hyperparameters are tuned for the best loss on imitating the teacher's output. Each student model is trained for 10 epochs on the dataset created by the output and derivatives of the teacher network. We used more than 1 epoch to ensure convergence of the models. We remark that the errors on the solution are calculated with respect to the NCL model, as no numerical solver we tried converged to the true solution.

**Results.** As Equation (A2.D) is guaranteed by NCL design, table 14 reports results for Equations (A2.M) and (A2.I). Although minimal, distillation reported an improvement in the BC (from 0.09 for the teacher to around 0.07 for the students). DERL and SOB outperform OUTL, with almost 50% error reduction in each metric. Even with a different architecture like NCL, these results highlight the effectiveness of derivative distillation for the transfer of physical knowledge across models.

Figure 19 shows the comparison among methodologies in terms of error differences between OUTL, SOB, and DERL so that positive blue regions are where our methodology performs better. We plot the errors on $\hat{u}_S, \mathrm{D}\hat{u}_S$ with respect to the NCL teacher model on the $Z = 0$ plane, as well as the PDE residual error on the momentum (A2.M) and incompressibility (A2.I) equations in table 8. Except for some initial conditions, DERL outperforms the other two methodologies, especially in the $\boldsymbol{u}, \mathrm{D}\boldsymbol{u}$ errors and in the momentum equation consistency.

### D.3 Transfer to a Larger Timespan

**Setup.** The timespan extension task we conducted shows how DERL can be leveraged to improve an existing physical model by extending its working time domain. DERL allows for the transfer of physical information from a teacher model to a student one that is currently training to solve an equation on new times. The equation of interest is the KdV equation (KdV) in table 8. The architecture of each model consists of 9 layers with 50 units each and tanh activation. In this case, no tuning is performed and the parameters are $\lambda_{\mathrm{BC}} = 0.001, \lambda_{\mathrm{IC}} = 10, \lambda_{\mathrm{PDE}} = 1.$, which provided optimal results with the PINN trained on the whole domain. We simulate a setting where the solution is to be found with no prior knowledge, as we are interested in the model that best works out-of-the-box. The full time-space domain is $[0, 1] \times [-1, 1]$,

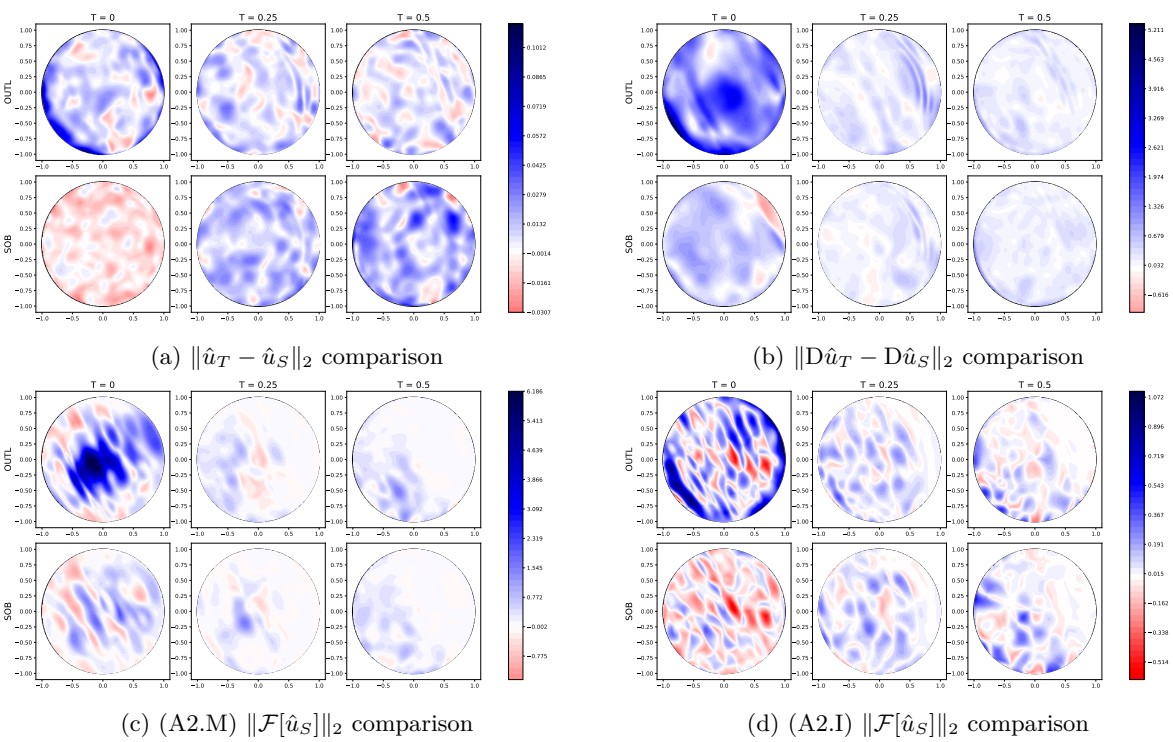

(a) $\|\hat{u}_T - \hat{u}_S\|_2$ comparison

(b) $\|D\hat{u}_T - D\hat{u}_S\|_2$ comparison

(c) (A2.M) $\|\mathcal{F}[\hat{u}_S]\|_2$ comparison

(d) (A2.I) $\|\mathcal{F}[\hat{u}_S]\|_2$ comparison

Figure 19: Neural Conservation Laws distillation. Comparisons between the methodologies on $u, Du$ errors w.r.t. the reference model and PDE residuals on the $Z = 0$ plane. All plots represent the difference between the method error and the DERL error so that positive (blue) regions are where DERL performs better.

which we divide in three time intervals: the first $[0, 0.35] \times [-1, 1]$, the middle interval $[0.35, 0.7] \times [-1, 1]$ and the final $[0.35, 1] \times [-1, 1]$. The training procedures consists of three main steps and here we provide an in-depth description of them.

1. A PINN is trained for 30.000 steps with 1000 random collocations points each with the Adam optimizer in the first interval $[0, 0.35] \times [-1, 1]$. Once the PINN is trained, we evaluate its output $\hat{u}$ and partial derivatives $D\hat{u}$ (calculated via AD (Baydin et al., 2018)) on the collocation points and save them in a dataset.

2. A second model is trained for 30.000 steps and the Adam optimizer with the same loss as in equation 7 that consists of (a) a PINN loss term to find the solution to the equation in the second interval $[0.35, 0.7] \times [-1, 1]$, (b) a distillation loss on the dataset from step 1, that is on points of the region $[0, 0.35] \times [-1, 1]$. This specific loss depends on the method used for distillation, which can be either DERL, OUTL, or SOB. (c) The initial condition loss (c). The boundary condition loss for the new extended region $[0, 0.7] \times [-1, 1]$. In this step, we consider two possibilities: we load the weights from the model of step 1, which we indicate with *continual*, or we start from a randomly initialized network, which we indicate with *from-scratch*. Finally, we consider a model that loads the weights of step 1 but has no distillation term for the lower-left quadrant and should forget that region: we call this model PINN with *no distillation*. Either way, we consider the final model to be a predictor for the solution $u$ in the cumulative interval $[0, 0.7] \times [-1, 1]$. We again save a dataset of its evaluation on the collocation points of the lower half of the domain.

3. We repeat step 2, this time by distilling information from $[0, 0.7] \times [-1, 1]$ and using the PINN loss on the remaining interval $[0.7, 1] \times [-1, 1]$, together with the BC loss on the whole region and the IC loss. We keep the *continual* and *from-scratch* pipelines separated: a model that starts from scratch in stage 2 will start from scratch again, and vice versa, a model that was fine-tuned in stage 2 will be

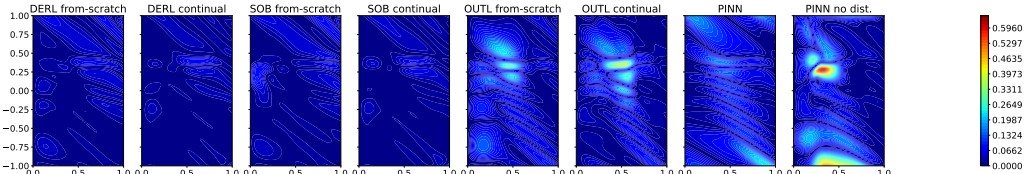

Figure 20: Prediction error in the time-space domain for the transferring experiment to larger timespans

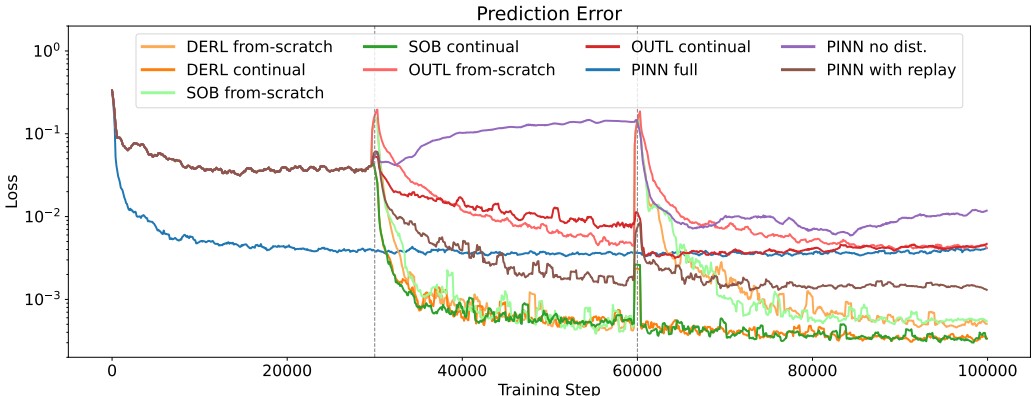

Figure 21: Prediction errors for the KdV transfer experiment. Vertical lines represent when a new step of incremental training is initiated.

fine-tuned in this stage as well. The final model is used to predict the solution in the entire domain $[0, 1] \times [-1, 1]$.

For a final comparison, we considered a PINN that is trained at once for the total sum of 100.000 steps in the full domain $[0, 1] \times [-1, 1]$, as this represents a model that is trained not incrementally.

**Results.** We now provide additional results for the transferring experiment on larger timespans in Section 5.1. Figure 20 shows the prediction error in the full time-space domain $(t, x) \in [0, 1] \times [-1, 1]$. We clearly see how DERL outperforms the model, especially compared to OUTL and the standard PINN, which converges early to a suboptimal solution. Finally, Figure 21 shows the prediction errors on the whole domain as a function of the training steps for all the methods. We see that the results of *from-scratch* are similar to the ones of the *continual* approach, with the latter performing slightly better.

### D.4 Transfer to new domain

**Setup.** The domain extension task we conducted shows how DERL can be leveraged to improve an existing physical model by extending its working domain. DERL allows for the transfer of physical information from a teacher model to a student one that is currently training to solve an equation in a new region of the domain. This way, the student model should be a good predictor of the solution of the equation in a larger domain than its teacher. The equation of interest is the Allen-Cahn equation (AC) in table 8. The architecture of the model is the same as in Section 4.1, that is we always work with MLPs with 4 layers of 50 neurons, similar to (Raissi et al., 2019). In this case, no tuning is performed. We simulate a setting where the solution is to be found with no prior knowledge, as we are interested in the model that best works out-of-the-box, without any particular training procedure. The full domain is $[-1, 1] \times [-1, 1]$, which we divide in the lower-left quadrant $[-1, 0] \times [-1, 0]$, the lower-right quadrant $[0, 1] \times [-1, 0]$ and the upper section $[-1, 1] \times [0, 1]$ as in figure 1. We first sampled a total of 1000 random collocation points in the full domain. We also calculated the

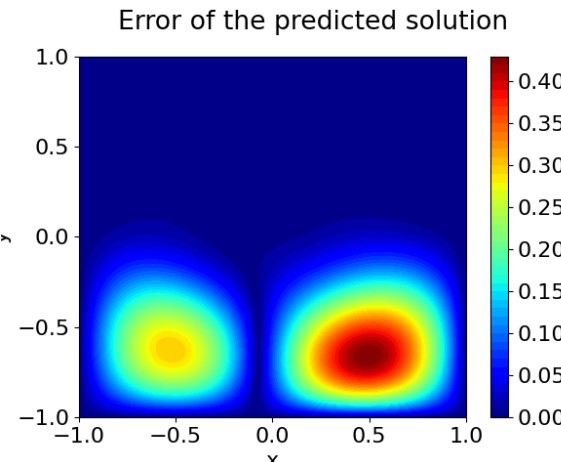

Figure 22: Error on $u$ for the PINN model with no distillation.

boundary conditions for each sub-region and the full one. Our experiment consists of three main steps and here we provide an in-depth description of them.

1. A PINN is trained for 100 epochs with the BFGS optimizer in the lower-left quadrant with the corresponding random collocation points and boundary conditions. Once the PINN is trained, we evaluate its output $\hat{u}$ and partial derivatives $D\hat{u}$ (calculated via AD (Baydin et al., 2018)) on the collocation points and save them in a dataset.

2. A second model is trained for 100 epochs and the BFGS optimizer with the same loss as in equation 7 that consists of (a) a PINN loss term to find the solution to the equation in the lower-right quadrant $[0, 1] \times [-1, 0]$, (b) a distillation loss on the dataset from step 1, that is on points of the region $[-1, 0] \times [-1, 0]$. This specific loss depends on the method used for distillation which can be either DERL, OUTL, or SOB. (c) The boundary condition loss for the new extended region $[-1, 1] \times [-1, 0]$. In this step, we consider two possibilities: we load the weights from the model of step 1, which we indicate with *continual*, or we start from a randomly initialized network, which we indicate with *from-scratch*. Finally, we consider a model that loads the weights of step 1 but has no distillation term for the lower-left quadrant and should forget that region: we call this model PINN with *no distillation*. Either way, we consider the final model to be a predictor for the solution $u$ in the lower half of the domain. We again save a dataset of its evaluation on the collocation points of the lower half of the domain.

3. We repeat step 2, this time by distilling the lower half of the domain $[-1, 1] \times [-1, 0]$ and using the PINN loss on the upper half of the domain $[-1, 1] \times [0, 1]$, together with the BC loss on the whole region. We keep the *continual* and *from-scratch* pipelines separated: a model that starts from scratch in stage 2 will start from scratch again, and vice versa, a model that was fine-tuned in stage 2 will be fine-tuned in this stage as well. The final model is used to predict the solution in the entire domain $[-1, 1] \times [-1, 1]$.

For a final comparison, we considered a PINN that is trained at once for the total sum of 300 epochs in the full domain $[-1, 1] \times [-1, 1]$, as this represents a model that is trained not incrementally.

**Results.** As seen in Section 5.2, the PINN model with no distillation performs poorly and forgets the information from the lower half of the domain, which can be noticed from the error in figure 22. We plot the error comparison of the other methodologies on $u$ and their PDE residuals in figures 23 and 24.

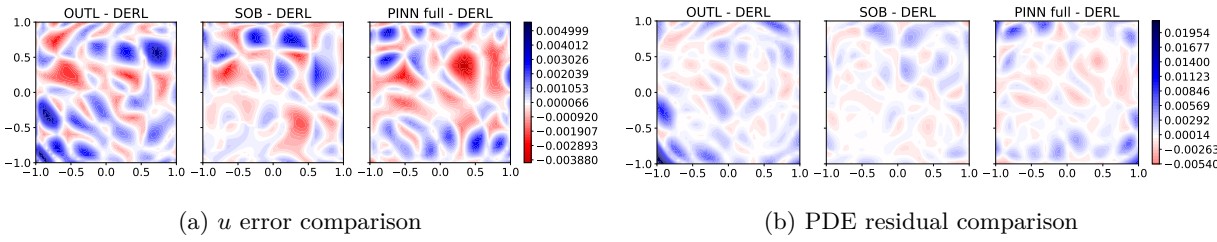

(a) $u$ error comparison

(b) PDE residual comparison

Figure 23: Comparison between DERL and the other models for the domain extension experiment with fine-tuning.

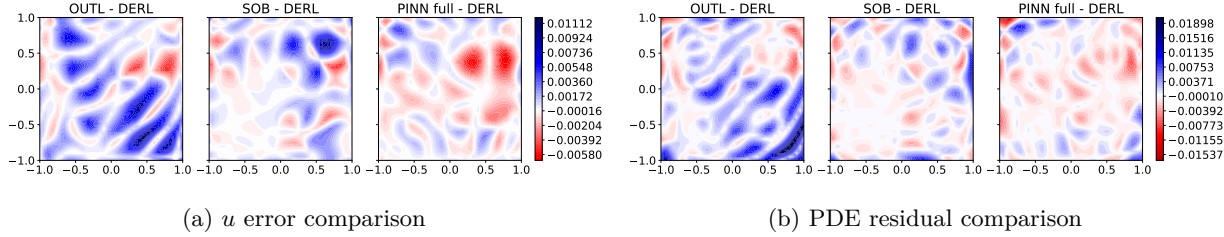

(a) $u$ error comparison

(b) PDE residual comparison

Figure 24: Comparison between DERL and the other models for the domain extension experiment starting from scratch at each step.

### D.5 Transfer to new parameters

**Setup.** We consider the Allen-Cahn equation (AC) in table 8 in its parametric form with two parameters. The parametric component is again $f$, which is calculated from the analytical solution

$$u(x, y; \xi_1, \xi_2) = \sum_{j=1}^{2} \xi_j \frac{\sin(j\pi x)\sin(j\pi y)}{j^2}$$

. The architecture of the model is the same as in Section 4.1, that is we always work with MLPs with 4 layers of 50 neurons, similarly to Raissi et al. (2019). The models take as input $(x, y, \xi_1, \xi_2)$ and predict $\hat{u}(x, y; \xi_1, \xi_2)$. In this case, no tuning is performed. We simulate a setting where the solution is to be found with no prior knowledge, as we are interested in the model that best works out-of-the-box, without any particular procedure. The domain is $[-1, 1] \times [-1, 1]$, and we consider a total of 25 couples of parameters $(\xi_1, \xi_2)$: 10 are used in the first step, 10 in the second step, and the rest for final testing purposes. We sample a total of 1000 random collocation points which will be used in every step, together with all couples of parameters. Our experiment consists of two main steps:

1. A PINN is trained for 100 epochs with the Adam optimizer and another 100 with the BFGS optimizer for all the combinations of the collocation points and the first 10 values of $\xi_1, \xi_2$. Once the PINN is trained, we evaluate its output $\hat{u}$ and partial derivatives $D\hat{u}$ (calculated via AD (Baydin et al., 2018)) on the collocation points and save them in a dataset.

2. A second model is trained for 100 epochs with the Adam optimizer and another 100 with the BFGS optimizer with a loss that consists of (a) a PINN loss term to find the solution to the parametric equation for the next 10 couples of $\xi_1, \xi_2$, (b) a distillation loss on the dataset from step 1, that is on previous values of the parameters. This specific loss depends on the method used for distillation, which can be either DERL, OUTL, or SOB. (c) The boundary condition loss. We again consider two possibilities: we load the weights from the model of step 1, which we indicate with *continual*, or we start from a randomly initialized network, which we indicate with *from-scratch*. Other possibilities are as in D.4. We finally test the models on the last 5 couple of parameters.

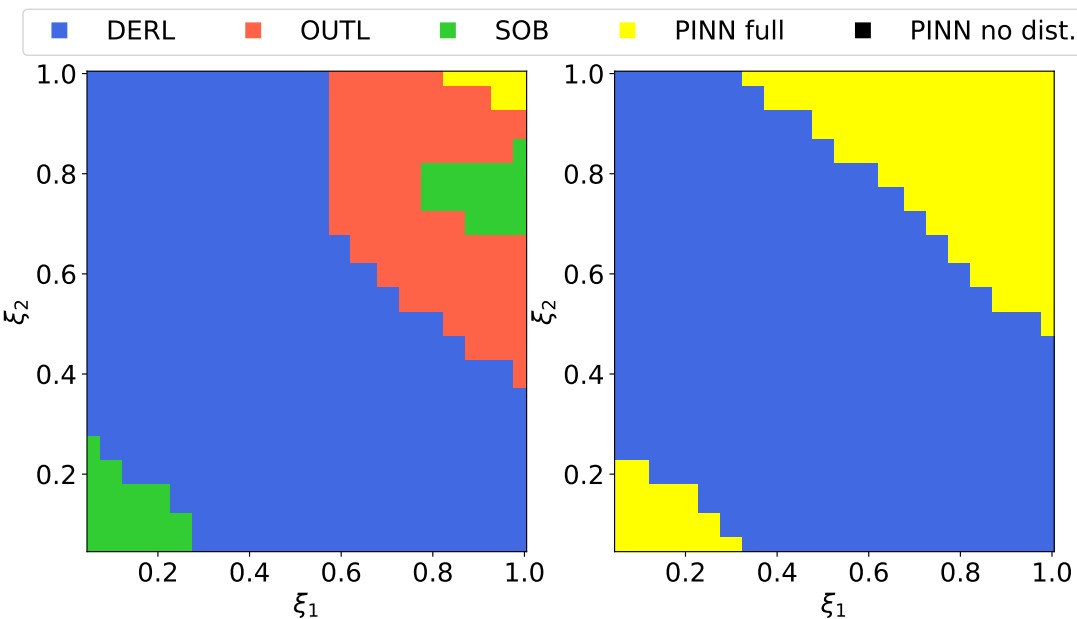

Figure 25: Allen-Cahn parameter transfer experiment with distillation on a new model from scratch. Each square represents the best model with the lowest error or PDE residual in the $\xi_1, \xi_2$ parameter space.

**Results.** In this case, the PINN with no distillation does not fall behind too much as it seems to maintain good generalization capabilities among the parameters.

## E  On the Effect of Noise and Step Size on DERL

In this Section, we experiment with empirical derivatives with different values for the step size $h$ and with white noise applied to the true derivative values, to simulate the effects of real measurements on DERL. We aim to show the robustness of DERL under realistic conditions. However, step size is not an issue in our experimental setup: as shown in Section 4.1 and in Appendix C.4.3, one can apply a cubic interpolation on the available data to calculate empirical derivatives with any step size. Furthermore, step sizes and noise are never an issue in the transfer learning setup of Section 5, where partial derivatives are always calculated on the teacher network, without any need for approximation.

We experiment with the Allen-Cahn equation in Section 4.1 for the in-domain generalization, and with the damped Pendulum experiment in Section 4.2 for the generalization to new parameters. We experiment with the step sizes $h = 0.001, 0.01, 0.05$ and with Gaussian noise with $\sigma = 0.001, 0.01, 0.05$. We also report the results of SOB and OUTL+PINN (only for the noise, as empirical derivatives are not used) with the same setup for a direct comparison of the two methods.

Results for the Allen-Cahn experiment are available in Table 15, while results for the damped pendulum experiment are available in Table 16. As we can see, DERL is robust to different noise values and with increasing step size. We notice that similar patterns observed in DERL are also present with SOB and OUTL+PINN, which is the least robust by far, showing that DERL extracts all the available information from partial derivatives, even without accessing information on $u$ directly.

Table 15: Results with different step size $h$ and noise $\sigma$ for the Allen-Cahn equation. We include the $L^2$ error on the prediction $u$ and the $L^2$ PDE residual.

| Model | $\|u - \hat{u}\|_2 \times 10^{-3}$ | $\|\mathcal{F}[\hat{u}]\|_2 \times 10^{-3}$ |
|---|---|---|
| *Varying step size $h$* | | |
| DERL $h = 0.001$ | 0.7654 | 2.097 |
| SOB $h = 0.001$ | 0.8733 | 2.173 |
| DERL $h = 0.01$ | 0.6769 | 2.071 |
| SOB $h = 0.01$ | 1.034 | 2.556 |
| DERL $h = 0.05$ | 4.904 | 4.226 |
| SOB $h = 0.05$ | 4.504 | 4.339 |
| *Varying noise $\sigma$* | | |
| DERL $\sigma = 0.001$ | 0.6250 | 1.870 |
| SOB $\sigma = 0.001$ | 1.497 | 3.589 |
| OUTL+PINN $h = 0.001$ | 9.176 | 14.97 |
| DERL $\sigma = 0.01$ | 1.398 | 2.430 |
| SOB $\sigma = 0.01$ | 1.854 | 4.007 |
| OUTL+PINN $h = 0.01$ | 8.537 | 13.45 |
| DERL $\sigma = 0.05$ | 3.726 | 6.321 |
| SOB $\sigma = 0.05$ | 4.235 | 5.717 |
| OUTL+PINN $h = 0.05$ | 19.06 | 15.57 |

Table 16: Results with different step size $h$ and noise $\sigma$ for the damped pendulum experiments equation. We include the same metrics as in Section 4.2 for the testing trajectories.

| Model | $\|u - \hat{u}\|_2$ | $\|\dot{u} - \hat{\dot{u}}\|_2$ | $\|\mathcal{F}[\hat{u}]\|_2$ | $\|\mathcal{G}[\hat{u}]\|_2$ | $\|\mathcal{F}[\hat{u}]_{t=0}^{\text{full}}\|_2$ |
|---|---|---|---|---|---|
| *Varying step size $h$* | | | | | |
| DERL $h = 0.001$ | 0.01237 | 0.01504 | 0.01581 | 0.1106 | 0.4678 |
| SOB $h = 0.001$ | 0.01670 | 0.02894 | 0.03018 | 0.1240 | 0.6704 |
| DERL $h = 0.01$ | 0.02002 | 0.01445 | 0.02161 | 0.07980 | 0.3565 |
| SOB $h = 0.01$ | 0.01726 | 0.02912 | 0.02919 | 0.1245 | 0.6627 |
| DERL $h = 0.05$ | 0.0574 | 0.06421 | 0.07272 | 0.2617 | 0.9187 |
| SOB $h = 0.05$ | 0.02927 | 0.04066 | 0.04015 | 0.1396 | 0.6785 |
| *Varying noise $\sigma$* | | | | | |
| DERL $\sigma = 0.001$ | 0.01536 | 0.01540 | 0.01678 | 0.09847 | 0.4440 |
| SOB $\sigma = 0.001$ | 0.008728 | 0.03116 | 0.02898 | 0.1482 | 0.6346 |
| OUTL+PINN $h = 0.001$ | 0.009781 | 0.05475 | 0.05515 | 0.2272 | 0.7717 |
| DERL $\sigma = 0.01$ | 0.02486 | 0.04159 | 0.03421 | 0.1872 | 0.5949 |
| SOB $\sigma = 0.01$ | 0.02203 | 0.04326 | 0.04245 | 0.2063 | 0.8317 |
| OUTL+PINN $h = 0.01$ | 0.02317 | 0.07735 | 0.07343 | 0.3511 | 0.8831 |
| DERL $\sigma = 0.05$ | 0.05432 | 0.09773 | 0.09407 | 0.4848 | 0.7811 |
| SOB $\sigma = 0.05$ | 0.04768 | 0.1098 | 0.1073 | 0.7025 | 1.652 |
| OUTL+PINN $h = 0.05$ | 0.05277 | 0.1848 | 0.1769 | 1.054 | 1.540 |

