# OpenReview forum: "Learning and Transferring Physical Models through Derivatives"
_TMLR — Accepted by TMLR_

### Review · Reviewer_khjG · 2025-10-28

**Summary Of Contributions:**

This paper proposed a framework for using neural networks to learn the evolution of physical systems, represented by differential equations. Rather than traditional methods, the proposed method, DERL, is trained on partial derivatives and can therefore achieve better performance in both theoretical domains, and is validated by empirical experiments. Besides, generalizability is also showcased via distillation for the DERL model. Experiments show that compared with other methods, the DERL framework can learn the model with higher accuracy and can be generalized well to unseen domains.

**Audience:**

Yes

**Audience Explanation:**

The topic of this work is using learning-based methods to solve complex differential equations, which is pervasive in the real world. The complexity of the real-world dynamical system naturally creates challenges for learning, and the results can, inversely provide insights for the machine learning community.

**Broader Impact Concerns:**

No ethical concern.

**Claims And Evidence:**

Yes

**Claims Explanation:**

The authors provide theoretical analysis about why learning derivatives can provide enough information for learning the solution of the differential equations. Apart from that, several empirical experiments are carried out and according to the reported results, the error terms relative to the baselines are indeed lower, which implies that the framework of DERL can achieve better performance than the baselines, including supervised learning and the PINN method. Therefore, the claim made by the authors is established.

**Requested Changes:**

1. In this work, one important part is the establishment of the cost function. In equation (6) the derivative information is used to perform supervised learning. However, in the real world, especially in a system that is dominated by an unknown DE, such a derivative "evaluation" can be hard to obtain. This is clearly a limitation of the work since, currently, the derivative is calculated by differentiation from rollouts. Although the authors mentioned that in the appendix section C, it would be more rigorous to put it in the main text to clearify the contribution of this work.

2. It would be nice if the authors can discuss more about the comparison between the proposed DERL framework and PINN framework. Essentially, in the proposed method, the residuals are discarded from the cost function and the authors claimed that the derivative information alone is enough (together with boundaries) to recover the full solution. This is pretty counterintuitive. Although the authors invoked the results about the Sobolev space, it would be clearer to see such a discussion explicitly as remarks or lemmas.

3. Fig 5 is visually hard to parse since the dotted lines and the other lines are entangled. It would be better if the author can present the illustration in a clearer way.

---

> ### Author Response · Authors · 2025-11-26
> **Author Response**
>
> We thank the reviewer for taking the time to review our work and provide valuable feedback. The requested changes have been implemented in the revised version of the paper. We make a summary of the raised points here, one by one:
> 1. We moved the discussion in Appendix C to the main text, at the end of Section 3, to highlight this contribution.
> 2. We updated Theorem 3.3 in our paper, with two additional remarks with an in-depth discussion, and an example that can be found in Section 3. The main difference between DERL and the PINN framework is that the latter is unsupervised and directly incorporates the information on the underlying equation into the loss function through the PDE residual term $\|\mathcal{F}[\hat{u}]\|^2$. On the contrary, DERL never sees the PDE directly, but is trained with the values of the derivatives of $u$ through the Derivative learning term. However, Theorem 3.3 shows that learning the derivatives is enough to recover a solution $\hat{u}$ that satisfies the PDE, that is, for DERL $\mathcal{F}[\hat{u}]\rightarrow 0$. Hence, DERL learns to satisfy the PDE without having access to it. This can be clearly seen in our experimental results, where we often perform best compared to OUTL augmented with the PINN loss. Finally, learning only $u$ is not enough for consistency. We also include a specific example in the revised version of the paper, and we further remark on higher-order derivative terms, although DERL performs well with only the first-order ones.
> 3. We modified Figure 5 by only considering the incremental training models with a continual approach (fine-tuning the previous model), as they perform best and represent the main contribution in this sense. The original Figure has been moved to the appendix for completeness.

---

### Review · Reviewer_6y99 · 2025-11-17

**Summary Of Contributions:**

The paper introduces Derivative Learning (DERL), a supervised method for learning physical systems by training a neural network to match partial derivatives of the target function rather than its direct values. The central idea is that, for ODEs and PDEs, the evolution of a physical system is fully determined by its derivatives, along with the initial and boundary conditions.

Strength:
1. The author evaluated multiple ODEs and PDEs for experiments, e.g., Allen–Cahn, Navier-Stokes, etc. It provides evidence that the proposed method works for solving differential equations; however, it requires knowledge of the partial derivatives.
2. The proposed method can be transferred between equations with different parameters, which means the proposed method not only learns the parameter of the PDE itself, but also the underlying system. This is an interesting point.
3. The idea of using partial derivatives to train the model and solve the PDE is interesting (Even though it is not practical, see the weakness, please).

Weakness:
1. The primary concern I have is that the proposed method requires the partial derivatives as the input data, e.g., it is known before the equation is solved. From my limited knowledge, it is not feasible for many PDE systems. This means we often see the function u rather than its derivative when solving the equation.  For instance, when we solve the heat equation, we can observe the temperature of a point in the domain at a given time, but we do not know the derivatives of the temperature with respect to time at that point.
2. As the author mentioned in the experiment setting, the partial derivatives used for training the model are computed from the solution or approximated by the numerical method, e.g., finite difference. For a fair comparison, it would be necessary to have the baseline for the solution from the traditional numerical method directly. Also, the baseline for the conventional numerical methods should be the one with strong performance, e.g., the finite element method, since it does not require an additional step to compute the partial derivatives.
3. The different solution for the different PDEs seems to use different neural network architectures. It looks like the author reports results from the selected hyperparameters without explaining the selection protocol or the candidate settings.

**Audience:**

Yes

**Audience Explanation:**

Using neural networks to solve differential equations is an interesting topic; it is related to AI for science, which is a critical domain from my point of view.

**Claims And Evidence:**

No

**Claims Explanation:**

Please see my points in the weakness section above.

**Requested Changes:**

Please address the point in the weakness section, thanks.

---

> ### Author Response · Authors · 2025-11-26
> **Author Response**
>
> We thank the reviewer for their time and important feedback on our work. We are happy the reviewer found our work interesting and appreciated our experimental results and the fact that we address the problem of transferring physical constraints.
> We now address the specific weaknesses and questions raised by the reviewer. We already implemented the corresponding changes in the revised version of the paper.
> 1. We would like to remark that, in the general case, the minimal requirement to apply DERL is to know $u$. If only $u$ is known, empirical derivatives can be easily calculated, and these are then used to train DERL. For example, this was the setting of the continuity equation experiment in Section 4.1, where only values of $u$ were available. Even in this case, DERL performs very similarly to OUTL in terms of prediction and better in terms of physical consistency. This is done by extracting all the information only from $u$, demonstrating the strength of our approach. Furthermore, we stress that our continual learning experiments in Section 5 are not affected at this concern since we use a fully unsupervised setting for learning the physical system. Supervised distillation with DERL uses the outputs of the model trained on previous steps (the teacher) as soft targets, without requiring any external label.
> 2. We would like to remark that the comparison between numerical methods and machine learning ones is not entirely fair, as they serve very different purposes and have complementary strengths and weaknesses. Specifically:
>      - Classical methods calculate the solution on a specific grid, which cannot be changed after calculation. To predict the values of the solution for a point not in the grid, these methods require recalculating the entire solution or relying on approximations or interpolations. While they are sometimes faster at “training”, they are much slower at inference. In general, classical methods are also more precise and have specific theoretical guarantees for convergence when the grid becomes finer and finer.
>     - Machine learning methods are fast at inference, as they can be queried for any possible point in the domain without retraining. On the other hand, their capabilities are limited by the available data, and there are often limited convergence guarantees.
>
>      Furthermore, similarly to our work, in many relevant works on machine learning methods for physical systems (such as [1,2,3]), the solution obtained by classical methods is often considered as the ground truth to match (such that the resulting system can generalize beyond that data). Finally, we remark that the applicability of classical methods depends on the specific system. In some cases, finite differences fail to converge in a reasonable time or can produce instabilities (for example, for the KdV equation, a Fourier transform method is instead used).
>
> 3. We clarify the choice of the network architectures in the revised version of the paper. We specify that these architectures are fixed at the beginning of the experiments, without any particular selection process a posteriori. In particular, the configuration used in most of the experiments (4 layers of 50 neurons) is the most common in [1], a seminal work in Physics-Informed machine learning. For the KdV equation in the transfer and incremental training tasks, the PINN required more layers to converge properly, similar to [2]. We adopted 9 layers. Finally, for the transfer experiment on the NCL model, we used the same architecture as the original work in [3]. The selection process of other hyperparameters is described in Appendix A.
>
> [1] Raissi et al. Physics-informed neural networks: A deep learning framework for solving forward and inverse problems involving nonlinear partial differential equations, Journal of Computational Physics, 2018
> [2] Jagtap et al., Conservative physics-informed neural networks on discrete domains for conservation laws: Applications to forward and inverse problems, Computer Methods in Applied Mechanics and Engineering, 2020
> [3] Richter-Powell et al., Neural Conservation Laws: A Divergence-Free Perspective, NeurIPS 2022.

---

### Review · Reviewer_fGEP · 2025-11-24

**Summary Of Contributions:**

This manuscript proposes DERL, a supervised learning framework that leverages derivative information to model physical systems and introduces a derivative-based distillation protocol to enable incremental learning. The authors demonstrate that by distilling derivatives rather than just output values, the model can effectively transfer knowledge across different time horizons, spatial domains, and physical parameters.

**Audience:**

Yes

**Audience Explanation:**

Machine Learning (ML) techniques have found great success in modeling dynamical and physical systems, including Partial Differential Equations (PDEs). This topic has attracted a great number of researchers. The paper focus on a derivative learning framework that can model physical systems by learning their partial derivatives, which also outperforms state-of-the-art methods in generalizing an ODE to unseen initial conditions and a parametric PDE to unseen parameters. I believe that the TMLR's audience will be interested in the findings in this paper.

**Broader Impact Concerns:**

Physics-informed neural networks has become a promising tool in solving PDEs via deep learning. The paper extends the working domain of a PINN and to generalize better across different PDE parameterizations. The work also shows promising intersections with continual learning, as DERL’s distillation allowed us to build physical models incrementally, showing minimal to no forgetting of previous knowledge.

**Claims And Evidence:**

Yes

**Claims Explanation:**

The paper is well-organized and supported by extensive experimental validation across various ODE and PDE tasks, showing clear improvements over standard baselines.

**Requested Changes:**

1. Justification of Distillation vs. Experience Replay: The authors frame the problem as incremental learning using distillation. However, it is unclear why a distillation-based approach is necessary compared to a simple Experience Replay baseline. Since the analytical derivatives (or empirical ones) are readily available, a straightforward alternative would be to retain a small buffer of old data points and train the model jointly with the new data. The paper needs to justify why we should trust a 'Teacher' model's compressed knowledge over the 'Raw Data's' ground truth. A comparison with a replay-based baseline would significantly strengthen the motivation for the proposed distillation protocol.

2. Incomplete Robustness Analysis (Missing PINN Comparison): The proposed method's reliance on 'empirical derivatives' computed via finite differences raises concerns about its robustness, a critical issue for real-world applications where measurement data is often noisy. While I acknowledge the sensitivity analysis provided in Appendix F, it is insufficient as it only demonstrates DERL's performance degradation in isolation (or compared to SOB). Crucially, the analysis omits a comparison against a standard PINN baseline under the same noisy conditions. This comparison is essential: a PINN treats the PDE residual as a soft regularization term, which typically offers inherent robustness against high-frequency noise. In contrast, DERL's supervised framework fits the model's gradients directly to noisy derivative targets, potentially causing it to overfit to high-frequency artifacts (noise amplification). To properly assess the method's practical utility, the authors should provide a comparative study demonstrating whether DERL maintains its advantage over PINNs in noisy regimes.

3. Asymmetric Baseline Comparison in Incremental Learning: The authors compare Incremental DERL against Joint Training PINN ("PINN full"), but omit the critical "DERL full" baseline (Joint Training on the entire domain using the DERL loss). Since Section 4 has already established that the DERL loss formulation is superior to the PINN loss, the current comparison in Section 5 conflates the benefits of the loss function with the effects of the training strategy. Without a "DERL full" baseline, it is impossible to isolate the impact of the incremental strategy or explicitly quantify the performance trade-off (e.g., error accumulation) incurred by using distillation instead of joint training. The authors should include this baseline to disentangle these factors.

---

> ### Author Response · Authors · 2025-11-26
> **Author Response**
>
> We thank the reviewer for their positive feedback. In this response, we address the requested changes:
> 1. We would like to clarify our setting for the experiments in Section 5. For these experiments, we assume that the physical system is learned in an unsupervised manner, with no target labels available during training. Hence, we work with unsupervised models such as PINNs and NCL.
> During training on each task, the PINN learns the current physical system. At the same time, previous knowledge is preserved via distillation with DERL. Here, DERL is not used to learn the physical system. It is only used to distill the knowledge contained in the model from the previous task to the current model. The predictions of the previous model act as a soft target for the distillation. Therefore, the distillation targets used by DERL do not come from real-world labels but from the model itself.
> Performing experience replay in this setting is not straightforward. One option could be to train the PINN during each step with points sampled from the current task and with points sampled from the previous tasks. This was our best intuition on the reviewer’s request. During each step, we sample 20% of points from each previous task. We performed this additional experiment, which we added to the results in Table 6 and Figure 5 under ‘PINN replay’. The results indicate that our distillation approach using DERL surpasses Experience Replay.
> 2. While our sensitivity analysis shows that DERL is robust with respect to a noisy label setting, we think that a direct comparison with standard PINN models in the same setting is not possible. PINNs are unsupervised models and do not rely on target labels. PINNs only require sampling points from the problem domain. Adding noise to the sampled points would just return other points in the domain.
> Nonetheless, we can provide the requested comparison for the OUTL+PINN model, as OUTL is a supervised approach. We provide these additional results in Appendix F (Tables 15 and 16), together with the previous sensitivity analysis. The results show that the effect of noise on OUTL+PINN is similar to that on SOB and DERL, and OUTL+PINN performs worse with all noise levels, especially on the Allen-Cahn experiment and on the consistency metrics on the Pendulum one.
> 3. Similar to our answer to point 1, we would like to clarify the difference between our experimental sections 4 and 5. The objective of Section 4 is to show that DERL can learn a physical system. Since DERL is a supervised method, we consider a supervised setting where ground truth data is available, either coming from an analytical solution or a numerical one. Instead, Section 5 shows that DERL can also be employed to incrementally train existing unsupervised models (such as PINNs) without forgetting. In this sense, DERL is a plug-in tool for any physical model that allows to mitigate forgetting when learning continuously. The targets for DERL do not come from the real systems but from the teacher model (soft targets).
>
> We ensured that these points were clarified in the revised version of the paper with a specific remark at the beginning of Section 5. We remain available for further clarification.

---

### Author Response · Authors · 2025-11-26
**Summary of Changes**

We thank the reviewers for their feedback and comments, which were carefully used to improve the quality of our paper. We uploaded a revised version that includes all the required changes to address the concerns that have been raised. We highlight these changes with a blue color. Here is a summary of the main changes:
- We moved Appendix C to the main text in Section 3 (rev. khjG).
- We clarified some aspects of Theorem 3.3, including additional remarks and discussion on how DERL learns physical systems and the differences with the PINN setup (rev. khjG, 6y99).
- We included references for the model architectures (rev. 6y99).
- We added a new baseline for the incremental training experiments in Section 5, that is the PINN replay (rev. fGEP).
- We improved the clarity of the setup in this Section, particularly regarding the unsupervised setting. (rev. fGEP).
- We added a sensitivity analysis with respect to noise for the OUTL+PINN baseline in Appendix F (rev. fGEP).

---

### Decision · Action_Editor_NSYh · 2026-01-13

**Recommendation:** Accept with minor revision

**Additional Comments:**

One additional suggestion I have is on the phrasing of the contributions. The abstract and introduction include the following sentences.

> We believe this is the first attempt at building physical models incrementally in multiple stages.

> To the best of our knowledge, we are the first to propose an approach to transfer knowledge across physical models to
study their incremental design through distillation.

Even with these qualifiers, the claims remain quite broad and may be read as stronger than intended, particularly by readers in adjacent fields who may interpret “first” differently depending on their conventions and prior art. To reduce the risk of confusion or dispute, it might be better to soften the wording and avoid explicit “first” claims.

**Audience:**

Yes

**Audience Explanation:**

Modeling physical systems through machine learning is an important topic within the TMLR community. In particular, the sub-community of PINN should find the results of this paper relevant and interesting. The noted connection to continual learning is also appreciated.

**Claims And Evidence:**

Yes

**Claims Explanation:**

This paper introduces a supervised learning approach for modeling physical systems using partial-derivative information rather than output values. In addition, it proposes a distillation protocol for transferring knowledge across domains. The paper reports an extensive set of experiments, and the results support the effectiveness of the proposed method.

Most reviewers agreed that the paper’s claims are supported by accurate, convincing, and clearly presented evidence. In response to the initial reviews, the authors added additional results in the revision. One reviewer noted a remaining concern about the lack of numerical methods as an experimental baseline. Although including such baselines would further strengthen the paper, the current experimental results appear consistent with the stated claims and, in my view, this issue should not preclude acceptance. The authors are encouraged to add the comparison between numerical methods and machine learning methods (in the response to reviewer 6y99) to the camera-ready version.

---

> ### Author Response · Authors · 2026-01-22
>
> Dear Editor,
> We would like to thank you for your work and feedback, which we appreciate.
> We implemented the suggested changes in the camera-ready revision of the paper, and we provided the links to the code and the video presentation.
> Let us know if there are remaining action items.
>
> Best regards,
> The authors